# Investigation on the Ductility Capacity of Concrete Columns with High Strength Steel Reinforcement under Eccentric Loading

**DOI:** 10.3390/ma16124389

**Published:** 2023-06-14

**Authors:** Feiyan Zhang, Xiang Liu, Fang-Wen Ge, Chenxing Cui

**Affiliations:** 1School of Management Engineering, Zhejiang Guangsha Vocational and Technical University of Construction, Dongyang 322100, China; zfy78@zjgsdx.edu.cn; 2School of Civil Engineering, Fujian University of Technology, Fuzhou 350118, China; liuxiang@fjut.edu.cn; 3School of Civil Engineering, Central South University, Changsha 410075, China; cui-chx@csu.edu.cn

**Keywords:** high strength reinforcement, ductility, numerical model, experiment, eccentric compression

## Abstract

Ductility-based structural design is currently the mainstream method. In order to analyze the ductility performance of concrete columns with high-strength steel reinforcements under eccentric compression, corresponding experimental studies have been performed. Numerical models were established, and their reliability was verified. Based on the numerical models, the parameter analysis was carried out, where eccentricity, concrete strength, and reinforcement ratio were considered to systematically discuss the ductility of the concrete column section with high-strength steel reinforcement. The results show that the ductility of the section under eccentric compression increases with the strength of the concrete and eccentricity, and decreases with the reinforcement ratio. Finally, a simplified calculation formula capable of quantitatively evaluating the section ductility was proposed.

## 1. Introduction

In the 1950s, some scholars proposed the concept of ductility based on the nonlinear response analysis and research of structures, which summarized the seismic performance of structures or components beyond the elastic deformation stage [1]. In engineering structural design, it is necessary to consider structure strength and stiffness, as well as ductility. As the early 1980s, countries around the world have used high performance and strength steel bars of 400 MPa or higher for practical engineering, and have used them as main reinforcement steel bars. After that, 500 MPa and 600 MPa grade steel reinforcement were further established [2,3,4,5]. The types and guideline of high-strength steel reinforcement in developed countries are similar. In addition to strength, the ductility of steel bars has attracted attention [6,7,8,9,10]. Due to the design requirements, concrete columns have two stress modes of axial pressure and eccentric pressure when bearing vertical load [11,12]. In extreme cases such as earthquakes, the moment of the eccentric compression column will increase, which may cause column eccentric pressure failure. The eccentric pressure failure mainly includes two modes: the one is small eccentric pressure failure, and the other is large eccentric pressure failure, in which the small eccentric pressure failure is a typical brittle failure, while the large eccentric pressure failure is a typical ductile failure.

There are many researches on the eccentric compression performance of columns, for examples, Chen and Chen et al. [13] experimentally studied flexural ductility of eight concrete-encased steel composite columns, and they proposed a method to evaluate flexural ductility of test specimens; Galano and Vignoli [14] analyzed strength and ductility of HSC and SCC columns under eccentric load; the strength and ductility of reinforcement column with CFRP under axial load and moment were analyzed in the works [15,16].

High-strength reinforcement can reduce the amount of steel used [17]. There were lots of researches on the configuration of high-strength reinforced concrete members, in the works of Refs. [18,19,20,21,22,23], the bearing capacity and seismic performance of the concrete members with 500 MPa high-strength steel reinforcement were analyzed. In addition to 500 MPa grade steel bars, some scholars also paid attention to 600 MPa grade steel bars, for examples, Feng et al. [24] conducted an experimental research on the seismic performance of reinforced concrete beam column joints with 600 MPa grade steel reinforcement, and the results showed that the joints with 600 MPa grade reinforcement have good seismic performance. Shi et al. [25] conducted experimental and theoretical researches on the seismic performance of concrete columns with 600 MPa grade reinforcement on the condition of high axial compression ratios, and suggested setting the design strength of 600 MPa grade steel reinforcements at 560 MPa.

Researchers have paid a lot of attention to the bearing capacity and seismic performance of concrete member with high-strength steel reinforcement, while few people holding on to the ductility of eccentrically compressed components. The ductility of members under load is insufficiently studied, and the ductility problem has been emphasized in the eccentric compression of concrete columns. The Eurocode EN 1998-3, 2005, contains expressions for the computation of the chord rotation capacity of RC members, and this paper will be a supplement to the quantitative research on the ductility of high-strength reinforced concrete columns under eccentric compression.

In this paper, the ductility performance of six concrete columns with 500 MPa grade (the yield strength was 500 MPa) high strength steel reinforcement under eccentric loading was analyzed by the experimental test and the numerical models. The various influencing parameters including reinforcement ratio, eccentricity ratio and concrete strength on the ductility of the specimen were systematically studied, and a simplified calculation formula for evaluating the ductility of the concrete section with high strength steel reinforcement under eccentric compression is proposed.

## 2. Experimental Research

### 2.1. Test Design

In the experiment, six concrete columns with 500 MPa grade high strength steel reinforcement under eccentric loading were manufactured. the length of columns are 1700 mm, and the cross-section of columns are all 300 mm × 400 mm, and the thickness of the protective layer is 25 mm for each column. The specimens were designed according to the Chinese code [26]; the details of the properties of specimens can be found in Table 1.

The size of the test specimens and the configuration of the reinforcement are shown in Figure 1. To simulate eccentric conditions in an actual situation, corbels were set at the ends of each column. To prevent partial crushing of the corbels, three reinforcement nets were added to the corbels, using 8 mm HPB300 steel reinforcements, and the spacing of the reinforcement nets was 75 mm. The stirrup was made of 8 mm HRB400 steel reinforcement, with a distance of 75 mm in the area of corbels and 200 mm in the middle. The finished test specimens are shown in Figure 2.

A knife hinge was set up at the lower end of the column for simulating the boundary conditions of the hinged ends of the column under eccentric compression, and the actuating head at the upper end of the column was a ball hinge. A 40 mm steel bar was placed on the upper plate. The schematic diagram and the physical diagram of the loading are shown in Figure 3.

### 2.2. Loading and Measurement

The monotonic continuous loading strategy was adopted, and the load is from zero until the specimen fails. By estimating the yield load, the loading process was divided into two stages. The first stage was before reaching 80% of the yield load, which was controlled by the force. Each step of the first stage has a load of 50 kN and the loading rate was 100 kN/min. Subsequently, there was the second stage, and the displacement control was adopted, and each step was 1 mm of this stage. The load would be stopped when the load drops to 65% of the ultimate bearing capacity.

The main measurement contents in the experiment include deflection, force and strain. The horizontal deflections were measured by LVDTs, while the vertical deflections were measured by the instrument itself. The strains of the reinforcements were measured by using a bonded strain gauge. The layout situations of the LVDT and reinforcement strain gauge are shown in Figure 4.

### 2.3. Material Properties

The mechanical properties of the steel materials are shown in Table 2. The stress-strain curve of reinforcement is shown in Figure 5. The 28-day standard compression strength of concrete of C50 and C70 is 53.81 MPa and 72.73 MPa, respectively.

## 3. Results of Tests and Analysis

### 3.1. General Behavior of Specimens

At the beginning of loading of each component, the load was small, no cracks appeared in the test specimens, the deflection and the strain of the steel bar and concrete increased linearly with the load, and the specimens showed elastic deflection characteristics. As the load increases, cracks began to appear on the tensile side of the specimens and began to widen and extend.

For the specimens HRCC-1, HRCC-4, HRCC-5, and HRCC-6, when the load was close to the ultimate load, the tensile steel bar began to yield, and the cracks developed rapidly. The crack width of the tensioned surface reached 1 mm. Subsequently, vertical cracks appeared, and the concrete was crushed. The primary crushing area was at the mid-span section of the specimen, and the bearing capacity reached the maximum. Under continuation of the loading, the vertical cracks in the concrete on the side near the load quickly spread and penetrated, and the concrete protective layer was peeled off. When the load dropped to 65% of the ultimate load, the load was stopped, and each specimen has significant deflection.

According to the failure form of the eccentric compression column, these specimens were damaged by large eccentric compression. The final failure pictures of the specimens are shown in Figure 6. It can be observed from the figure that the specimens had a large deflection, the concrete was peeled, and the cracks were more obvious. The compressed longitudinal steel bars were exposed, and the cracks in the mid-span area were triangular. It can be seen that the tensile steel reinforcement was yield before the concrete in the compression zone crushed, which was consistent with the general eccentric compression failure mode of ordinary reinforced concrete.

For specimens HRCC-2 and HRCC-3, the crack width reached 1.5 mm, and the maximum length was about 25 cm when the load approached the ultimate capacity. The distance between cracks was about 8 cm. Subsequently, the concrete at the edge of the compression zone was suddenly crushed, and the central crushing zone was concentrated in the middle of the specimen. The load reached the ultimate bearing capacity and quickly dropped below 80% of the ultimate bearing capacity, and the loading was ended. The final failure pictures of the specimens are shown in Figure 6, it can be observed that the deflection of the test piece was reasonable, the cracks were less developed, the concrete was spalled a little, and no apparent triangle zone failure formed. It can be known that the strain of the concrete at the edge of the compression zone reached the limit compressive strain before the tensile steel bar yields. According to the eccentricity judgment conditions, it can be judged that the specimens were damaged by small eccentric compression. The ductility of the test specimen was low, and no apparent signs appeared before failure.

### 3.2. Load-Deflection Curves

The load-horizontal deflection curves at the midpoint of column of each specimen are shown in Figure 7. It can be understood that the columns were at the elastic stage before cracking. The deflection and load had a linear relationship, and the initial stiffness of each column was the same; after the concrete on the tension side cracks, the slope of the load-deflection curve became smaller and increased nonlinearly; when the columns reached the ultimate bearing capacity, there were slight decline of bearing capacity, and then the load-deflection curves of HRCC-1, HRCC-4, HRCC-5, and HRCC-6 slowly decreased, and the bearing capacity of HRCC-2 was directly dipped to below 80% of the ultimate load (There is also a reason here that there was an instrument malfunction during the loading process).

It can be found that under different reinforcement ratio parameters, HRCC-2 had a 21% increase in bearing capacity compared to HRCC-1, the deflection corresponding to the ultimate vertical bearing capacity increased by 23%, and the displacement decreased by 8% when the bearing capacity dropped to 85%.

Compared with HRCC-5, the bearing capacity of HRCC-6 increased by 31%, the displacement corresponding to the ultimate bearing capacity increased by 16%, and the displacement decreased by 29% when the bearing capacity dropped to 85%. The reinforcement ratio had a greater effect on the bearing capacity, and the bearing capacity increases with the reinforcement ratio. After the specimen cracked, the deflection of the specimen with a large reinforcement ratio increased rapidly, and the specimens with a small reinforcement ratio under the same load had a larger deflection. After the HRCC-1 and HRCC-5 reached the ultimate bearing capacity, the applied load decreased more gently, and the ductility was better than that of the specimen with a large reinforcement ratio.

Under different concrete strength conditions, the bearing capacity of HRCC-1 was 0.4% different from HRCC-3, and HRCC-6 was 7% higher than HRCC-4. It can be seen that the concrete strength within the test parameters had no effect on the bearing capacity, while after the specimens cracking, the specimens with higher concrete strength had greater stiffness than that with lower concrete strength.

Under different eccentricity conditions, the bearing capacity of HRCC-5 was 22% lower than that of HRCC-1, and HRCC-6 was 16% lower than HRCC-2. It can be seen that within the range of the experiment parameters, the eccentricity ratio had a great influence on the specimens, and the bearing capacity decreased with the increase of the eccentricity ratio. This is because the specimens with a large eccentricity had a smaller depth of compression zone, which reduces the vertical load-bearing capacity. After cracking, the specimens with large eccentricity had less rigidity and larger lateral deflection under the same load. This was because the concrete with large eccentricity cracks faster in the compression zone, which leads to faster deflection.

### 3.3. Reinforcement Strain

Whether the tensile steel reinforcement yield first during failure can be regarded as the judgment basis for eccentric compression. Figure 8 is the load-strain relationship curve of the tensile and compressive steel bars during the loading process of each specimen, where the ordinate is the measured longitudinal load during loading, and the abscissa is the resistance strain gauge arranged at the mid-span. The measured strain in the steel reinforcement is positive under tension and negative under compression. It can be seen that at the initial stage of loading, the strains of the tensile and compressive steel bars increase linearly with the increase of the load, and the deflection state of the test piece also shows elastic characteristics. As the concrete in the tension zone cracked and no longer beared the tensile force, the depth of the concrete compression zone decreased, and all the tensile force was beared by the longitudinal tensile steel reinforcement, and the slope of the load-strain curve of the tensile steel reinforcements became smaller. Except HRCC-2, the tensile steel bars of the other test pieces showed that the steel bars began to yield when loaded to the ultimate bearing capacity. When the limit load is reached, the concrete in the compression zone is crushed, and the load-reinforcement strain curve begins to decline. From the load-strain situation, except for HRCC-2, the failure modes of the other specimens are all eccentric compression failures.

## 4. Numerical Analyses

Due to the limited experimental research, numerical models were created to make systematic evaluations for concrete columns with high strength steel reinforcement under eccentric loading. The sectional numerical models were established by using XTRACT [27] software, and the method adopted by the software is to mesh the section, as shown in Figure 9, then gradually increase the sectional curvature, and use the numerical integration method to obtain the corresponding moment, which has high calculation accuracy and efficiency, and the disadvantage is that the influence of column length, i.e., slenderness ratio, cannot be considered.

### 4.1. Model Establishment and Validation

The stress-strain relationship of concrete adopts the default unconstrained Mander model. The 28-day standard value of compressive strength of concrete cylinders is related to the standard value of compressive strength *R_s_* of concrete cubes used in China’s concrete code as *f_c_*′ = 0.85 *R_s_*. The strength of the concrete material corresponding to the model in the test are given in Table 3. The stress-strain relationship of steel adopts the parabolic strain hardening steel model, which is also called two-line model considering reinforcement phase.

Based on the above material stress-strain relationship, the same numerical analysis model as the test parameters is used to calculate the loading process. The moment and curvature of the specimen can not be directly obtained during loading, where the moment was obtained by the load and eccentricity, and the curvature was converted from the data collected by the strain gauge pasted on the reinforcement, and it can be written as,
(1)$$\phi =\frac{\left(\varepsilon_{y}+\varepsilon_{y}{}^{\prime }\right)}{h-2a_{s}}$$
where $\varepsilon_{y}$ and $\varepsilon_{y}{}^{\prime }$ denote strain of tensile and compressive steel reinforcement in the midspan of the column; h denotes the height of column section, $a_{s}$ denotes the distance from the concrete surface to the center of the longitudinal reinforcement.

The moment-curvature curve obtained from the numerical calculation results is compared with the test results, as shown in Figure 10.

Since the curvatures were obtained through strain conversion, and strain testing is not stable, the experimental moment curvature curves of some specimens, such as HRCC-2 and HRCC-5, are not ideal. HRCC-3 experienced instrument failure during the loading process of the test, the curve after the bearing capacity decreased failed to captured. However, the test curve before reaching the ultimate bearing capacity was close to the numerical curve, and the numerical results could compensate for the shortcomings of the test. However, overall, the numerical results curves were consistent with the experimental results curves.

### 4.2. Parametric Analysis

In order to more accurately analyze the eccentric pressure performance of HRCC columns, XTRACT was used in a parameter analysis. 102 kinds of section of concrete columns with high strength steel reinforcement under eccentric loading with different concrete strengths, reinforcement ratio, and eccentricities were designed. The concrete strength grades were C25, C30, C35, C40, C45, C50, C55, C60, C65, C70 total 10 cases; There were 9 types of reinforcement ratios: 0.34%, 0.42%, 0.52%, 0.63%, 0.71%, 0.82%, 0.95%, 1.05% and 1.23%; The eccentricity had a total of eight cases of 0.475, 0.525, 0.575, 0.625, 0.6975, 0.75, 0.8 and 0.85. The dimensions of each section were the same as those in the test and are all 300 mm × 400 mm. The ductility factor was calculated the following formula,
(2)$$\mu_{\phi }=\frac{\phi_{u}}{\phi_{y}}$$
where $\phi_{u}$ denotes the ultimate curvature, usually, this value can be considered as the corresponding ultimate curvature when the bearing capacity decreases to 85% of the ultimate bearing capacity; *φ_y_* denotes the yield curvature, for components with obvious yield points, this value is relatively easy to determine, but for components without obvious yield points, some measures need to be taken, for examples, Colajanni et al. [28] and Palanci [29] proposed corresponding equations by studying the effects of cross-sectional and material parameters on moment curvature response, as well as by extensively evaluating the contribution of parameters assigned to the prediction equation through statistical analysis. In this paper, the yield curvature was determined by the method shown in Figure 11: firstly, using the initial stiffness as an auxiliary line OA, and line OA intersects with the horizontal line of the ultimate load at point A; secondly, creating a vertical line AB, and line AB intersects with the bending moment curvature curve at point C; thirdly, connecting point O and point C, and extending it, and the line intersects with the horizontal line of the ultimate load at point D; lastly, the curvature corresponding to point D is the yield curvature.

#### 4.2.1. Influence of Concrete Strength

Figure 12 is the moment-curvature curve corresponding to different concrete strengths in the case of four combinations of eccentricity and reinforcement ratio. The strength of concrete has little effect on the initial stiffness of the specimen. As the curvature increases, the load of the specimen with small concrete strength rises slowly, and the ultimate moment capacity of the section increases with the strength of the concrete. With the same moment, the section with small concrete strength has a greater curvature than that with large concrete strength; when the moment reaches the ultimate bearing capacity, the curve of sections with high concrete strength is smoother. Within a certain range, the concrete columns with higher strength concrete exhibit better ductility.

Based on the analytical results of concrete parameters, the influence of different concrete strengths on the ductility capacity of concrete columns with high strength steel reinforcement under eccentric loading with different eccentricity and reinforcement ratio is shown in Figure 13 where the ordinate is the section curvature ductility coefficient, and the abscissa is the standard value of concrete strength. It can be seen that the effect of concrete strength on the ductility of the section is not significant under the condition of a large reinforcement ratio (reinforcement ratio equals to 1.23%), and the section ductility coefficient increases with the increase of the concrete strength when the reinforcement ratio is small (reinforcement ratio equals to 0.82%), especially in the case of large eccentricity (eccentricity equals to 0.750). This is because the reinforcement ratio was too large, resulting in the section being in a small eccentric compression failure state.

Overall, the strength of concrete had increased from 20 MPa to 45 MPa, the ductility of cases exhibits varying degrees of increase, in which the maximum growth rate reached 61%. It can be concluded that in the design of section of concrete columns with high strength steel reinforcement under eccentric loading, adjusting the strength of the concrete relative to adjusting the reinforcement ratio has no significant effect on the ductility of the section. In the case of proper reinforcement ratio and eccentricity, the ductility of sections can be improved by properly increasing the concrete strength.

#### 4.2.2. Effect of Reinforcement Ratio

Figure 14 is the section moment-curvature curves corresponding to different reinforcement ratios under the case of four eccentricity ratio and concrete strength combinations. The reinforcement ratio has a greater influence on the capacity of the section of the eccentric compression column, and the specimen with a higher reinforcement ratio had a larger bearing capacity. At the beginning of loading, the reinforcement ratio has no obvious effect on the stiffness of the specimen. After the edge of the tensile zone was cracked, the slopes of the moment-curvature curve of the section become smaller. Under the same action of the moment, the cross-section curvature of the specimen with a small reinforcement ratio is larger. When the section with smaller reinforcement ratio reaches the ultimate bearing capacity, its bending moment-curvature curve performs more gentle, which means that the sections with smaller reinforcement ratios have better ductility.

Figure 15 shows the influence of the reinforcement ratio on the ductility of the sections with the case of eccentricity equals to 0.625 and 0.750, and the concrete strength grades were C50 and C70. It can be seen that in four cases when the reinforcement ratio reaches 1.23%, the section ductility reaches about 2.0, which shows brittle failure. The cross-section ductility of the section decreases as the reinforcement ratio increases, and for sections with higher concrete strength, the reduction amplitude was larger. It can also be known that when the other conditions were the same, the ductility of the sections with higher concrete strength was generally larger, and the ductility of the sections with larger eccentricity was generally larger, for small eccentricity, when the reinforcement ratio increases from 0.34% to 1.23%, the ductility decreases by about 58%, and for large eccentricity, the ductility decreases by about 70%.

#### 4.2.3. Effect of Eccentricity Ratio

Figure 16 is the moment-curvature curve of the mid-span section of each section obtained by numerical analysis. Before the concrete cracking, the eccentricity has little effect on the section stiffness, and after the concrete cracking, the slope of the section moment-curvature curve decreases, and the reduction amplitude increases with the increase of the eccentricity. After reaching the ultimate bearing capacity, the greater the section eccentricity, the smoother the bending moment curvature curve and the better the ductility.

Figure 17 shows the influence of eccentricity on the curvature ductility section. It can be seen that the ductility of the column section is more sensitive to the reinforcement ratio than to concrete. In addition, In the four cases, the section ductility increases with the increase of eccentricity, the change is greater when the reinforcement ratio is smaller. For a smaller reinforcement ratio, when the eccentricity increases from 0.475 to 0.85, the section ductility increases by about 113%, and for a larger reinforcement ratio, the ductility increases by about 45%.

### 4.3. Ductility Calculation

According to the previous analysis, it can be concluded that ductility capacity of concrete columns with high strength steel reinforcement under eccentric loading is related to the longitudinal reinforcement ratio, concrete strength, and eccentricity ratio. The sensitivity of ductility to each parameter is in the order of reinforcement ratio, eccentricity ratio and concrete. The section ductility decreases with the increase of reinforcement ratio, increases with the increase of eccentricity, and increases with the increase of concrete strength. Regression analysis is performed based on the results of the section ductility coefficient obtained from the numerical analysis, and the mathematical expression is as follows,
(3)$$\left\{\begin{array}{l} \mu =\frac{1}{0.85\xi \;+\;0.05}\left(\xi \leq 0.5\right)\\ \mu =2.0(\xi >0.5) \end{array}\right.$$
where *ξ* is the relative depth of compression zone, and it can be calculated by the following formula,
(4)$$\xi =\frac{x}{h_{0}}$$
where $h_{0}$ denotes the effective depth of section; *x* denotes the depth of compression zone, and when the section is subjected to large eccentric compression failure, it can be calculated by the following formula,
(5a)$$\left\{\begin{array}{l} N=\alpha_{1}f_{c}bx+f_{y}^{\prime }A_{s}^{\prime }-f_{y}A_{s}\\ Ne=\alpha_{1}f_{c}bx\left(h_{0}-\frac{x}{2}\right)+f_{y}^{\prime }A_{s}^{\prime }\left(h_{0}-a_{s}^{\prime }\right) \end{array}\right.$$ where $\alpha_{1}$ is a constant coefficient, $f_{c}$ denotes strength of concrete, b is the width of section, $f_{y}^{\prime }$ and $f_{y}$ are the yield strength of compression and tension reinforcement respectively, $A_{s}^{\prime }$ and $A_{s}$ denote sectional area of compression and tension reinforcement respectively, *e* is the eccentricity.

When the section is subjected to small eccentric compression failure, *x* can be calculated by the following formula,
(5b)$$\left\{\begin{array}{l} N=\alpha_{1}f_{c}bx+f_{y}^{\prime }A_{s}^{\prime }-\sigma_{s}^{}A_{s}^{}\\ Ne=\alpha_{1}f_{c}bx\left(h_{0}-\frac{x}{2}\right)+f_{y}^{\prime }A_{s}^{\prime }\left(h_{0}-a_{s}^{\prime }\right) \end{array}\right.$$ where $\sigma_{s}$ denotes the stress of tension reinforcement. More details can be found in the code GB50010-2010 [26].

The calculation result of the regression formula is shown in Figure 18. By using this fitting equation, the ductility coefficient of concrete columns with high strength steel reinforcement under eccentric loading can be quickly obtained.

## 5. Conclusions

The ductility capacity of concrete columns with high strength steel reinforcement under eccentric loading was investigated. The ductility was analyzed based on experiments, and the numerical model was estimated, and the parameter analysis was performed. Through extensive analysis of the results, the influence of various parameters on ductility was obtained, and a calculation method for ductility coefficient was proposed. Some conclusions can be drawn as follows,
(1)The failure modes of concrete columns with high strength steel reinforcement under eccentric loading are similar to that with ordinary steel reinforcement.(2)The ductility of the column section under eccentric compression increases with the increase of concrete strength, but in some cases there was no obvious pattern.(3)The ductility of column section decreases with the increase of reinforcement ratio, especially when the eccentricity is large, and the ductility of the sections with larger eccentricity is generally larger. For small eccentricity, when the reinforcement ratio increases from 0.34% to 1.23%, the ductility decreases by about 58%, and for large eccentricity, the ductility decreases by about 70%.(4)The ductility of column section increases with the increase of eccentricity ratio, and the change is greater when the reinforcement ratio is smaller. For a smaller reinforcement ratio, when the eccentricity increases from 0.475 to 0.85, the section ductility increases by about 113%, and for a larger reinforcement ratio, the ductility increases by about 45%.(5)The simplified calculation formula of curvature ductility can accurately evaluate the ductility of the concrete columns with high strength steel reinforcement under eccentric loading.(6)The ductility of the concrete columns with high strength steel reinforcement under eccentric loading can be improved by improving the concrete strength and increasing eccentricity ratio. Increasing the reinforcement ratio has a negative impact on ductility, so when there is a demand for high ductility, it is necessary to control the reinforcement ratio to not exceed a certain limit value.(7)There are some limitations in the work of this manuscript, for examples, the results were based on cross-sections and ignored the influence of slenderness ratio, and the bidirectional eccentricity was not considered. In future research, more attention will be paid to spatial effects.

## Figures and Tables

**Figure 1 materials-16-04389-f001:**
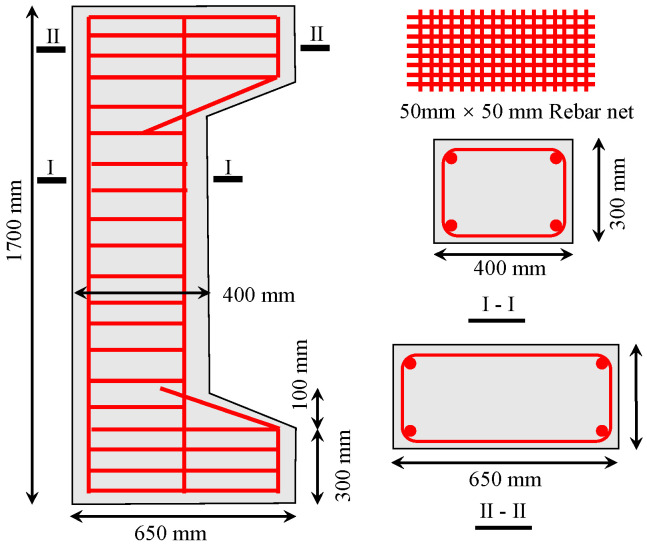
Schematic diagram of specimen.

**Figure 2 materials-16-04389-f002:**
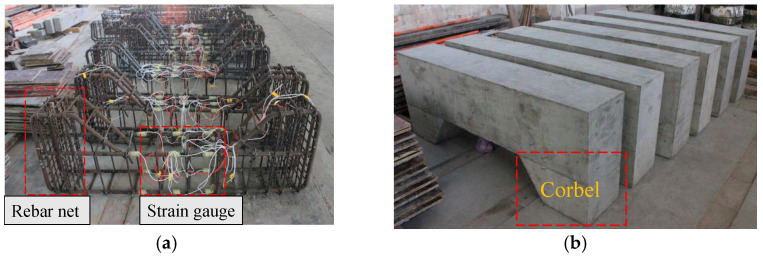
Specimens: (**a**) Reinforcement cage; (**b**) Finished specimens.

**Figure 3 materials-16-04389-f003:**
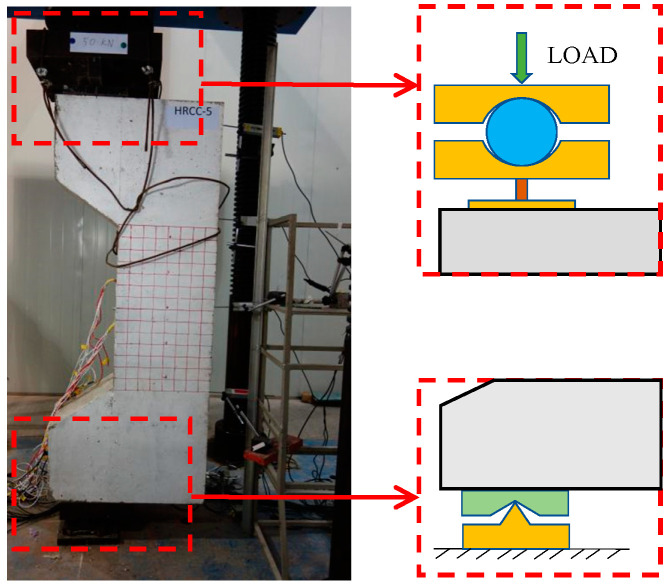
Specimen loading device.

**Figure 4 materials-16-04389-f004:**
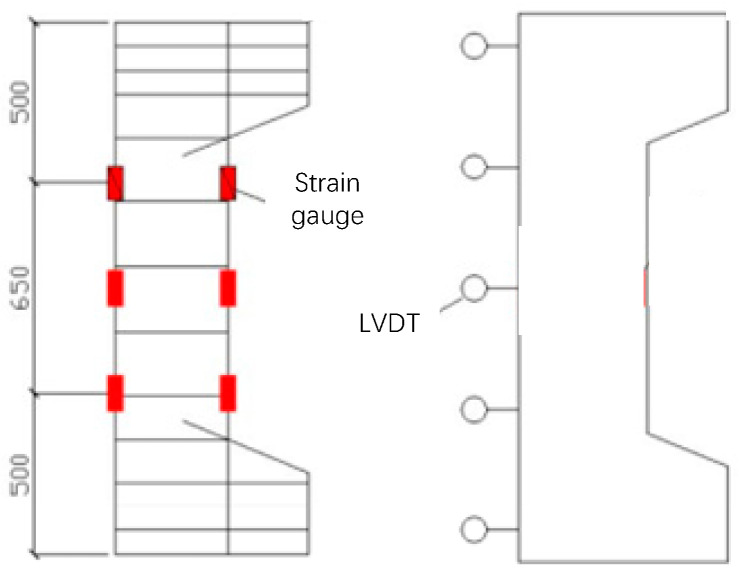
Arrangement of strain gauges and LVDTs.

**Figure 5 materials-16-04389-f005:**
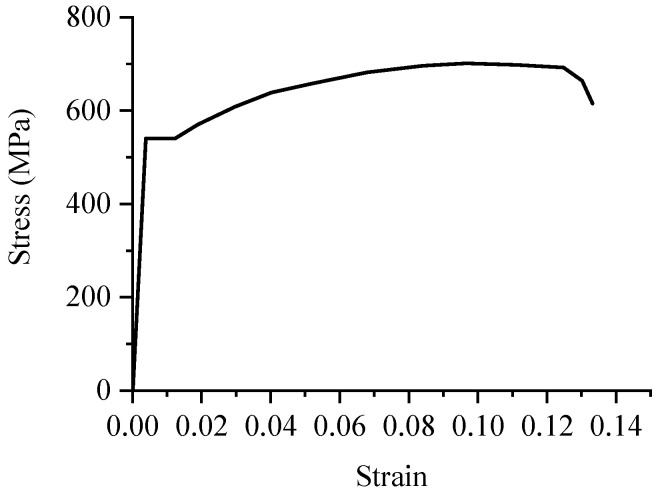
Strain-stress of reinforcement.

**Figure 6 materials-16-04389-f006:**
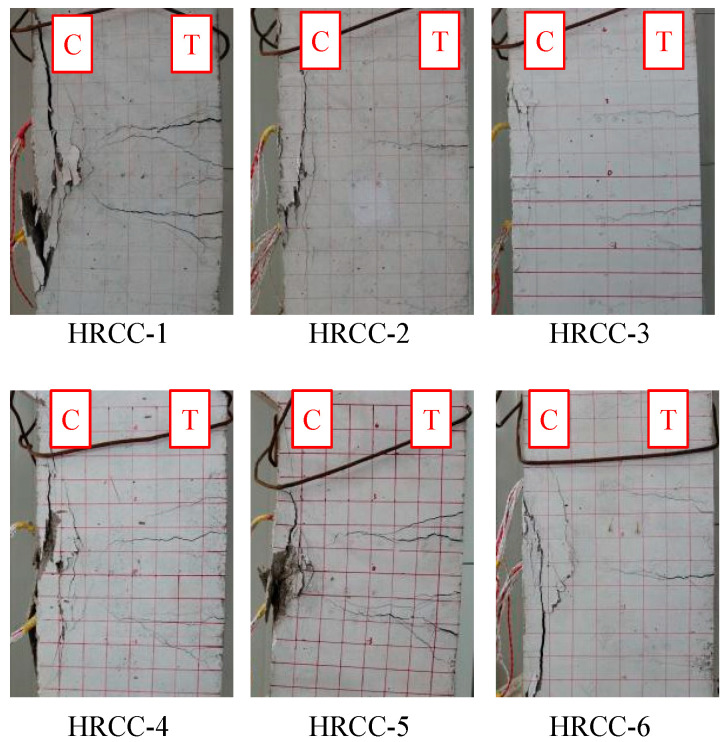
Damage image of specimens (C means compression side and T means tension side).

**Figure 7 materials-16-04389-f007:**
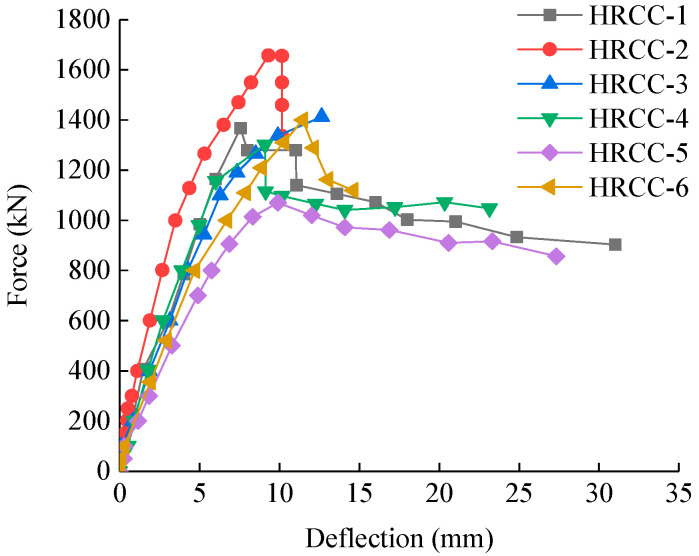
Load-deflection curves.

**Figure 8 materials-16-04389-f008:**
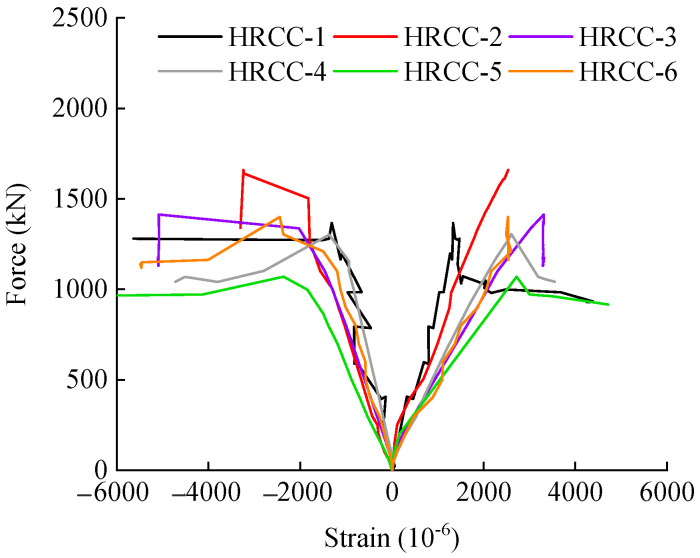
Load-strain curves of reinforcement.

**Figure 9 materials-16-04389-f009:**
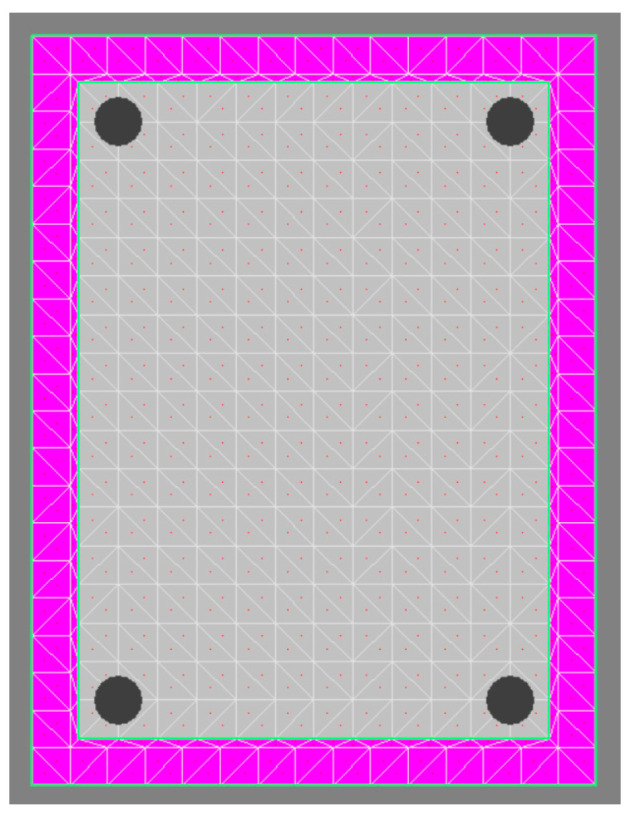
Fiber cross-section in XTRACT.

**Figure 10 materials-16-04389-f010:**
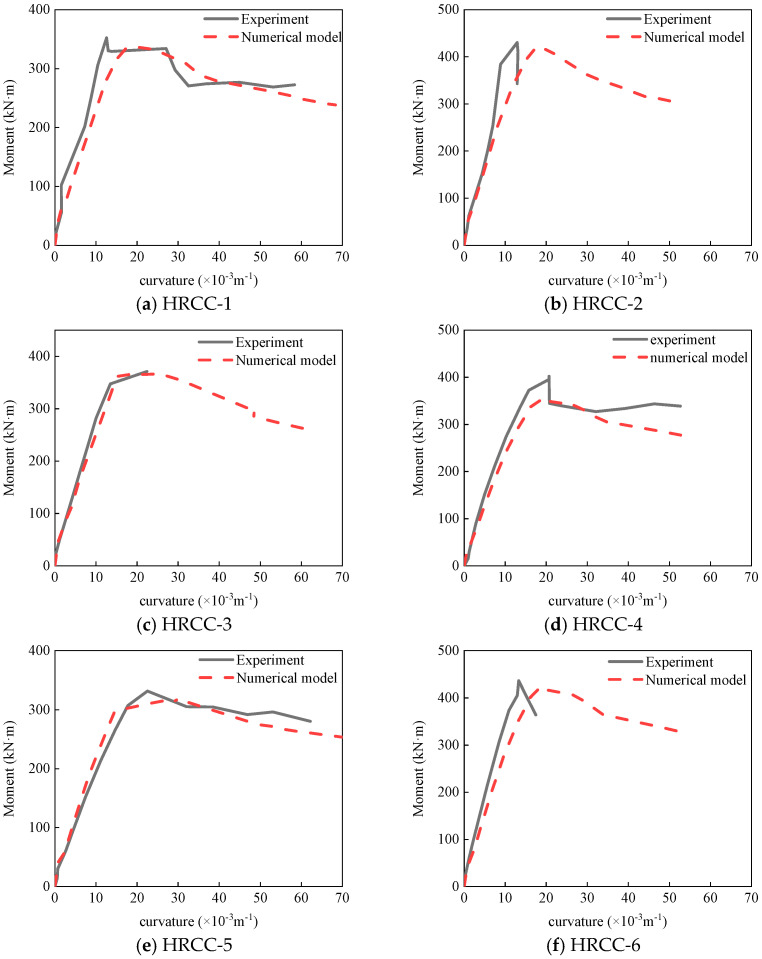
Comparison of test and numerical simulation.

**Figure 11 materials-16-04389-f011:**
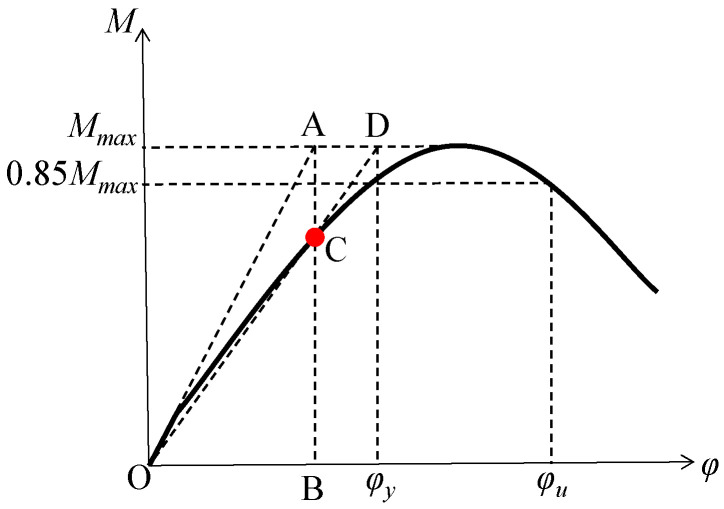
Determination of ultimate and yield curvature.

**Figure 12 materials-16-04389-f012:**
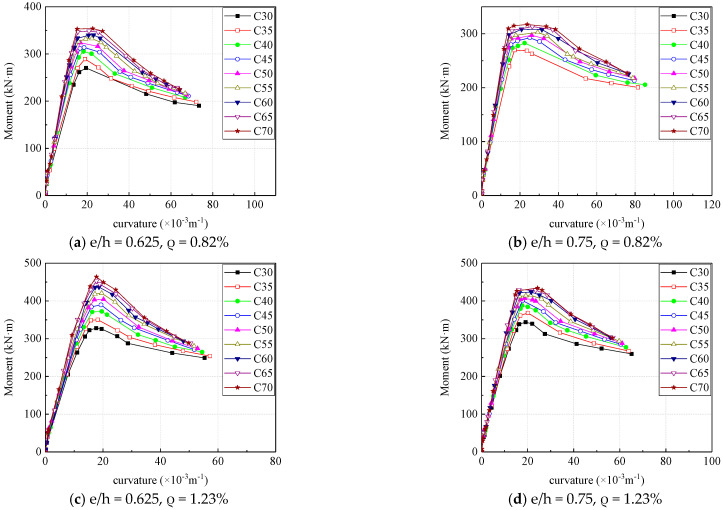
Effect of concrete strength on section moment-curvature curve.

**Figure 13 materials-16-04389-f013:**
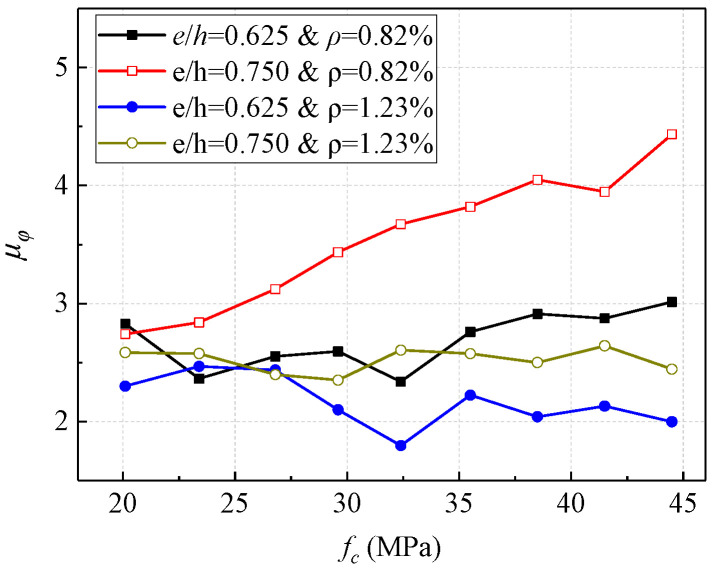
Influence of concrete strength on section ductility.

**Figure 14 materials-16-04389-f014:**
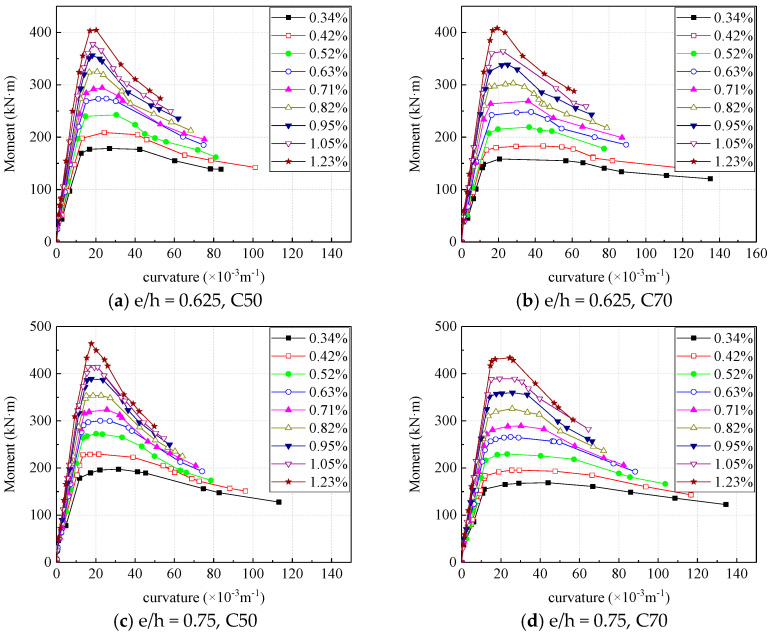
Effect of reinforcement ratio on moment-curvature curve.

**Figure 15 materials-16-04389-f015:**
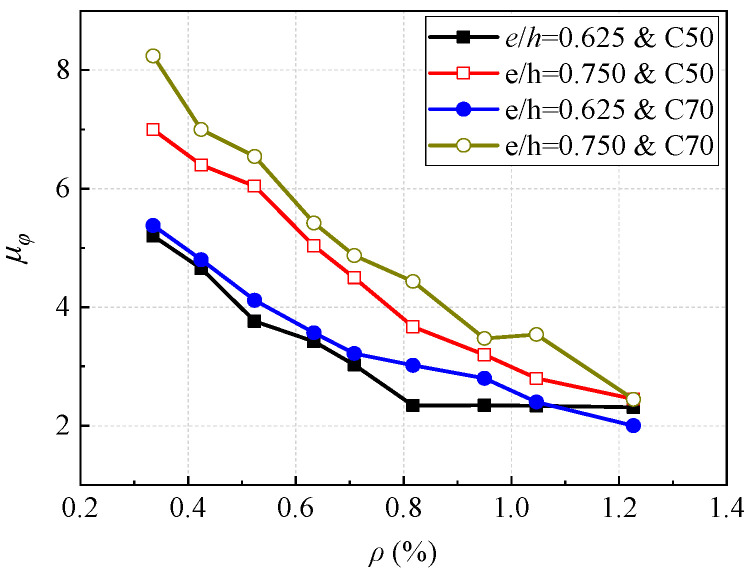
Effect of reinforcement ratio on section ductility.

**Figure 16 materials-16-04389-f016:**
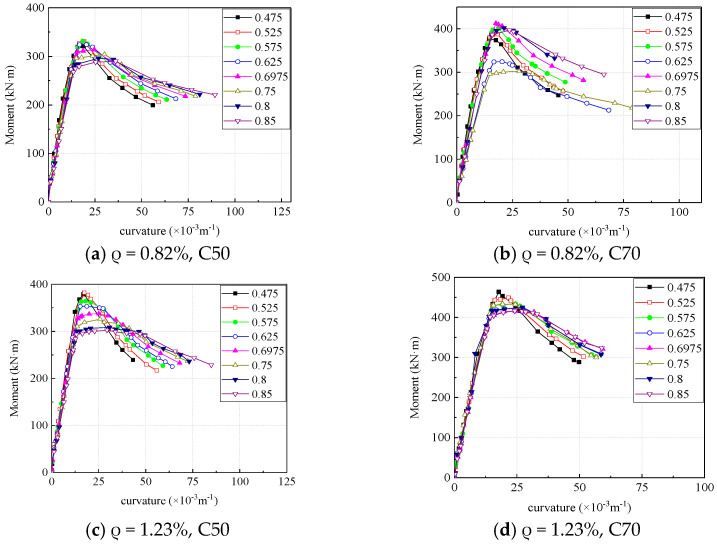
Effect of eccentricity on moment-curvature curve.

**Figure 17 materials-16-04389-f017:**
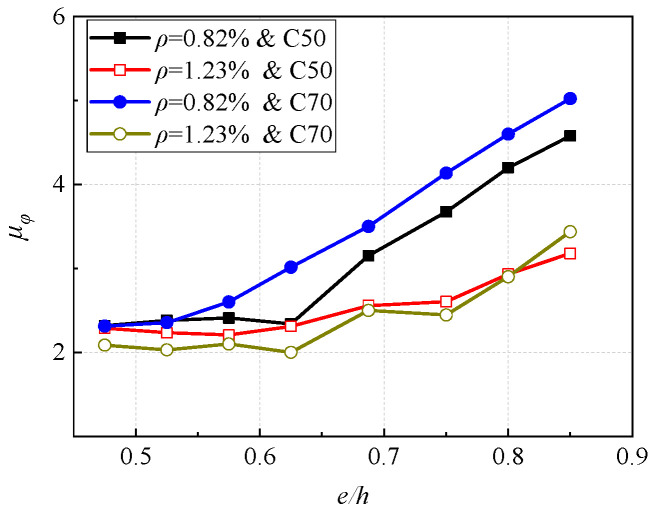
Effect of eccentricity on section ductility.

**Figure 18 materials-16-04389-f018:**
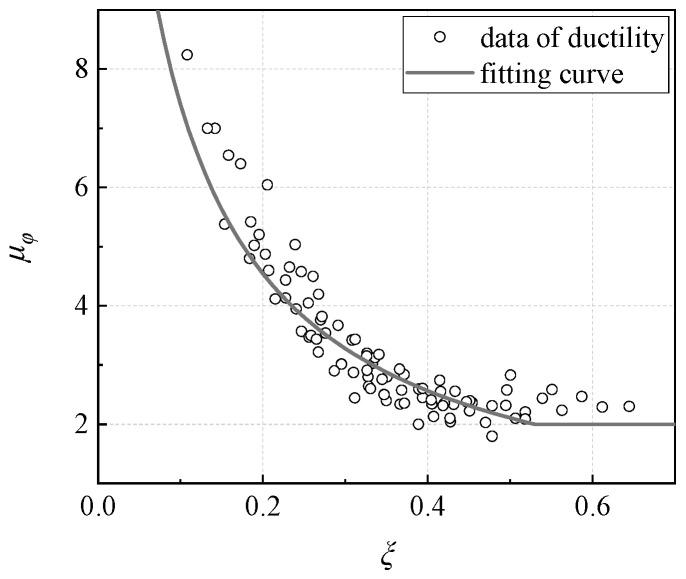
Ductility calculation.

**Table 1 materials-16-04389-t001:** Properties of specimens.

Specimen Number	Concrete Strength Grade	Longitudinal Reinforcement Ratio/%	Eccentricity/mm
HRCC-1	C50	0.82	250
HRCC-2	C50	1.23	250
HRCC-3	C70	0.82	250
HRCC-4	C70	1.23	300
HRCC-5	C50	0.82	300
HRCC-6	C50	1.23	300

**Table 2 materials-16-04389-t002:** Properties of experimental steel reinforcement.

Reinforcement Type	Diameter/mm	Yield Strength/MPa	Ultimate Strength/MPa
HRB500	25	540	675
HRB400	8	420	510

**Table 3 materials-16-04389-t003:** Model concrete strength value.

Concrete Strength Grade	*R_s_*/MPa	*f_c_*′/MPa
C50	53.81	45.74
C70	72.73	61.82

## Data Availability

Not applicable.

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
