# Peer review of "Investigation on the Ductility Capacity of Concrete Columns with High Strength Steel Reinforcement under Eccentric Loading"

_materials, 2023, doi:10.3390/ma16124389_

Round 1

Reviewer 1 Report

The study titled “Investigations on the ductility performance of concrete column under eccentric load with high strength steel rebar” applies some experimental studies for concrete columns with high strength steel bar and uses some numerical models to understand behavior and ductility. Content of study is suitable to Material journal. However, it needs comprehensive and major issues and revisions to be addressed. Some issues are given below for authors:

1.      The use of English is very weak. The language of manuscript should be considerably improved by native speaker. It is very hard to understand at times even for very known issues. Annotations for authors are given with attached file for authors. In addition to annotations, paper needs comprehensive language check!

2.      Title of the manuscript may be also revised. The title might be re-written as “Investigation on the ductility capacity of concrete columns with high strength steel rebar under eccentric loading”.

3.      Current version of manuscript is like a technical report describing the experimental results. In addition to experimental results, obtained results should be discussed with approving or contrary results by studies in the literature.

4.      Although authors used high steel strength bars, they are working on concrete columns with very known behavior. Accordingly, references of the manuscript should be considerably improved by considering studies in the literature.

5.      Page 10 is blank page. Are there any figures describing the experimental setup? Needs revision!

6.      Authors should also add some numerical values and discuss with literature describing the crushing strain of concrete, or buckling of longitudinal bars, or strain rates at different stages of experiment.

7.      Authors did not describe the strain values or any termination criterion for the determination of the ultimate curvature of member in parametric analysis section. How did authors decide for ultimate level or which criteria did they use to describe ultimate curvature? The rationale behind the ultimate curvature should be described. Because determination of ultimate values is critical for the ductility of sections.

8.      In addition, determination of yield curvature is not clear! How did authors calculate the yield curvature? Figure 9 is not clear to describe these levels (both ultimate and yield curvature)

9.      There must be erroneous in Figure 14 (especially for Fig. 14b and Fig 14d). Needs correction!

10.   In eq. 2 authors provided an equation to calculate ductility of RC section with high strength rebars and equation is dependent to relative compression zone parameter (ξ). However, they did not describe how to calculate this parameter. Needs clarification.

11.   Determination of only ductility is not adequate and has no important value especially in structural manner. On the other hand, determination of yield curvature (or estimation of this value) will increase value of proposed equation. Accordingly, authors are strongly recommended to quote following studies to emphasize that it will be possible to obtain ultimate curvature with their proposed equation if the yield curvature is determined (for beams or columns).

     A.     “Flexural Response Prediction of Reinforced Concrete Members Based on Statistical Observations” doi: https://doi.org/10.1007/s13369-016-2392-z

     B.     “Analytical prediction of ultimate moment and curvature of RC rectangular sections in compression” doi: https://doi.org/10.1007/s10518-013-9471-2

12.   The conclusion section of manuscript is superficial and needs improvement describing the novelty and improvements of current study for the literature and possible future studies.

13.   Study has very limited experimental specimens. Accordingly, limitations of the study should be also highlighted.

14.   Additional comments regarding typo errors and comments provided in the attached file.

The use of English is very weak. The language of manuscript should be considerably improved by native speaker.

Author Response

#1 Reviewer:

General comment: The study titled “Investigations on the ductility performance of concrete column under eccentric load with high strength steel rebar” applies some experimental studies for concrete columns with high strength steel bar and uses some numerical models to understand behavior and ductility. Content of study is suitable to Material journal. However, it needs comprehensive and major issues and revisions to be addressed. Some issues are given below for authors:

Response: Thank you very much for your comments. We have revised the manuscript, the revised part was marked in red in the revised manuscript, and the point-by-point reply are shown below, please check.

Comment 1: The use of English is very weak. The language of manuscript should be considerably improved by native speaker. It is very hard to understand at times even for very known issues. Annotations for authors are given with attached file for authors. In addition to annotations, paper needs comprehensive language check!

Response 1: Thank you for your review. The language of manuscript has been considerably improved by native speaker, and we have double checked the manuscript.

Comment 2: Title of the manuscript may be also revised. The title might be re-written as “Investigation on the ductility capacity of concrete columns with high strength steel rebar under eccentric loading”.

Response 2: Thank you for your review. We have revised the title, and the revised title is “Investigation on the ductility capacity of concrete columns with high strength steel rebar under eccentric loading”.

Comment 3: Current version of manuscript is like a technical report describing the experimental results. In addition to experimental results, obtained results should be discussed with approving or contrary results by studies in the literature.

Response 3: Thank you for your comment. We compared it with concrete columns with ordinary reinforced in terms of failure modes and basic mechanical properties:“It can be seen that the tensile steel reinforcement was yield before the concrete in the compression zone crushed, which was consistent with the general eccentric compression failure mode of ordinary reinforced concrete..”

Please see Lines 145-146 in the revised manuscript for more details.

Comment 4: Although authors used high steel strength bars, they are working on concrete columns with very known behavior. Accordingly, references of the manuscript should be considerably improved by considering studies in the literature.

Response 4: Thank you for your review. We have revised the introduction and add more references:

“High-strength reinforcement can reduce the amount of steel used [17]. There were lots of researches on the configuration of high-strength reinforced concrete members, in the works of Refs. [18–23], the bearing capacity and seismic performance of the concrete members with 500 MPa high-strength steel reinforcement were analyzed. In addition to 500MPa grade steel bars, some scholars were paying attention to 600MPa grade steel bars, for examples, Feng et al. [24] conducted experimental research on the seismic performance of reinforced concrete beam column joints with 600MPa grade steel reinforcement, and the results showed that the joints with 600MPa grade reinforcement have good seismic performance. Lee et al. [25] studied the performance of 600MPa plates and circular hollow section joints under external torque and shear forces through experiments and finite element analysis, and compared the results with current design equations. Lee et al. [26] studied the influences of microstructure and inclusion on cold cracking with 600 MPa grade steel welds. Shi et al. [27] conducted experimental and theoretical research on the seismic performance of concrete columns with 600MPa grade reinforcement on the condition of high axial compression ratios, and suggested setting the design strength of 600MPa grade steel reinforcements at 560MPa.”

Please see Lines 46-64 in the revised manuscript for more details.

Comment 5: Page 10 is blank page. Are there any figures describing the experimental setup? Needs revision!

Response 5: Thank you for your review. We have revised the manuscript.

Comment 6: Authors did not describe the strain values or any termination criterion for the determination of the ultimate curvature of member in parametric analysis section. How did authors decide for ultimate level or which criteria did they use to describe ultimate curvature? The rationale behind the ultimate curvature should be described. Because determination of ultimate values is critical for the ductility of sections.

Response 6: Thank you for your review. The reviewer provided a very constructive suggestion. In this manuscript, we used some relatively simple methods to determine the ultimate curvature and yield curvature. The ultimate curvature was considered to be the curvature corresponding to the reduction of bearing capacity to 85%, while the yield curvature was determined using the methods introduced in the manuscript. These can be found in the manuscript:

where  denotes the ultimate curvature, usually, this value can be considered as the corresponding ultimate curvature when the bearing capacity decreases to 85% of the ultimate bearing capacity; φy denotes the yield curvature, for components with obvious yield points, this value is relatively easy to determine, but for components without obvious yield points, some measures need to be taken, for examples, Colajanni et al. [30] and Palanci [31] proposed corresponding equations by studying the effects of cross-sectional and material parameters on moment curvature response, as well as by extensively evaluating the contribution of parameters assigned to the prediction equation through statistical analysis. In this paper, the yield curvature was determined by the method shown in Figure 10.

Please see Lines 271-285 in the revised manuscript for more details.

Comment 7: In addition, determination of yield curvature is not clear! How did authors calculate the yield curvature? Figure 9 is not clear to describe these levels (both ultimate and yield curvature)

Response 7: Thank you for your review. We have revised the manuscript: “the yield curvature was determined by the method shown in Figure 10: firstly, using the initial stiffness as an auxiliary line OA, and line OA intersects with the horizontal line of the ultimate load at point A; secondly, creating a vertical line AB, and line AB intersects with the bending moment curvature curve at point C; thirdly, connectting point O and point C, and extending it, and the line intersects with the horizontal line of the ultimate load at point D; lastly, the curvature corresponding to point D is the yield curvature.”

Please see Lines 279-285 in the revised manuscript for more details.

Comment 8: There must be erroneous in Figure 14 (especially for Fig. 14b and Fig 14d). Needs correction!

Response 8: Thank you for your carefully review. We have revised it. Please see Figure 14 in the revised manuscript for more details.

Comment 9: In eq. 2 authors provided an equation to calculate ductility of RC section with high strength rebars and equation is dependent to relative compression zone parameter (ξ). However, they did not describe how to calculate this parameter. Needs clarification.

Response 9: Thank you for your review. We have added more details about ξ in the revised manuscript.

Please see Lines 372-376 in the revised manuscript for more details.

Comment 10: Determination of only ductility is not adequate and has no important value especially in structural manner. On the other hand, determination of yield curvature (or estimation of this value) will increase value of proposed equation. Accordingly, authors are strongly recommended to quote following studies to emphasize that it will be possible to obtain ultimate curvature with their proposed equation if the yield curvature is determined (for beams or columns).

  1. “Flexural Recd Concrete Members Based on Statistical Observations” doi: https://doi.org/10.1007/s13369-016-2392-z
  2. “Analytical prediction of ultimate moment and curvature of RC rectangular sections in compression” doi: https://doi.org/10.1007/s10518-013-9471-2

Response 10: Thank you for your review. This is a very good opinion that has opened up some of our research ideas. The focus of our discussion in this manuscript was on the ductility, where yield curvature and ultimate curvature were obtained using relatively simple methods. We will pay more attention to this aspect in future research. We have added some papers in this area to the manuscript:

φy denotes the yield curvature, for components with obvious yield points, this value is relatively easy to determine, but for components without obvious yield points, some measures need to be taken, for examples, Colajanni et al. [30] and Palanci [31] proposed corresponding equations by studying the effects of cross-sectional and material parameters on moment curvature response, as well as by extensively evaluating the contribution of parameters assigned to the prediction equation through statistical analysis.”

References:

  1. Colajanni, P.; Fossetti, M.; Papia, M. Analytical Prediction of Ultimate Moment and Curvature of RC Rectangular Sections in Compression. Bull. Earthq. Eng. 2013, 11, 2055–2078, doi:10.1007/s10518-013-9471-2.
  2. Palanci, M. Flexural Response Prediction of Reinforced Concrete Members Based on Statistical Observations. Arab. J. Sci. Eng. 2017, 42, 3689–3709, doi:10.1007/s13369-016-2392-z.

Please see Lines 271-277 in the revised manuscript for more details.

Comment 11: The conclusion section of manuscript is superficial and needs improvement describing the novelty and improvements of current study for the literature and possible future studies.

Response 11: Thank you for your review. We have added the description of novelty and improvements of current study for the literature and possible future studies:

“There are some limitations in the work of this manuscript, for examples, the results were based on cross-sections and ignored the influence of slenderness ratio, and the bidirectional eccentricity was not considered. In future research, more attention will be paid to spatial effects.”

Please see Lines 411-414 in the revised manuscript for more details.

Comment 12: Study has very limited experimental specimens. Accordingly, limitations of the study should be also highlighted.

Response 12: Thank you for your review. The limitations of the study were added in the manuscript:

There are some limitations in the work of this manuscript, for examples, the results were based on cross-sections and ignored the influence of slenderness ratio, and the bidirectional eccentricity was not considered.

Please see Lines 411-414 in the revised manuscript for more details.

Comment 13: Additional comments regarding typo errors and comments provided in the attached file.

Response 13: Thank you for your review. We have revised the manuscript according to the PDF attached file.

Reviewer 2 Report

The authors presented a well established experimental and analytical study on the ductility of high-strength concrete columns with high-strength reinforcement. The topic is practical and scientifically sound. The reviewer noticed some issues in the manuscript that should be revised before the work is ready to be published.

In Table 1, the unit for reinforcement ratio should be %? Please clarify.

Please include more information about the high-strength reinforcement bar used in this study. A stress-strain curve compared with common commercial rebars would be helpful to justify the motivation and significance of the study. 

Figure 5, labeling the tension side and compression side would help. 

Line 118-121. The sentence is not clear. And also not clear why and how "the ductility is good", as stated by the author. It would be better to quantify the characteristic properties first and then discuss them. So it is recommended to reorganize the flow of section 3. 

Figure 7 needs a legend for the curves. Also, it's not very reasonable to combine different failure modes in one single graph,. It is recommended to separate the curves into two sub-figures. The "steels" in the figure caption is not clear to the reviewer. Also, it should be "load-deflection" rather than "load-strain". 

line 231. A blank page. 

Figure 8. It seems the result curves have been smoothed. That will deeply damage the credibility of the figure.  And the figure itself is not sufficient to show the "good predictability". A table with the experimental and simulated characteristic properties and the comparison will be much more convincing. 

Please also double-check the variable symbols. Some of them in the figures were not in italics.

Some of the sentences are hard to follow. It is recommended to have professional proofreading and revision on some of the wording, especially on the figure captions and discussions.  

Author Response

#2 Reviewer:

General comment: The authors presented a well established experimental and analytical study on the ductility of high-strength concrete columns with high-strength reinforcement. The topic is practical and scientifically sound. The reviewer noticed some issues in the manuscript that should be revised before the work is ready to be published.

Response: Thank you very much for your comments. We have revised the manuscript, the revised part was marked in red in the revised manuscript, and the point-by-point reply are shown below, please check.

Comment 1: In Table 1, the unit for reinforcement ratio should be %? Please clarify.

Response 1: Thank you very much for your careful review. We have revised it.

Comment 2: Please include more information about the high-strength reinforcement bar used in this study. A stress-strain curve compared with common commercial rebars would be helpful to justify the motivation and significance of the study. 

Response 2: Thank you very much for your comment. We have added the curve in the revised manuscript: “The stress-strain curve of reinforcement is shown in Figure 5.”

Please see Lines 118-119 in the revised manuscript for more details.

Comment 3: Figure 5, labeling the tension side and compression side would help.

Response 3: Thank you very much for your comment. We have added the labels of the tension side and compression side in Figure 6.

Comment 4: Line 118-121. The sentence is not clear. And also not clear why and how "the ductility is good", as stated by the author. It would be better to quantify the characteristic properties first and then discuss them. So it is recommended to reorganize the flow of section 3.

Response 4: Thank you very much for your comment. Yes, the ductility cannot be determined solely from experimental results, we have moved this sentence, and reorganized the flow of section 3.

Comment 5: Figure 7 needs a legend for the curves. Also, it's not very reasonable to combine different failure modes in one single graph,. It is recommended to separate the curves into two sub-figures. The "steels" in the figure caption is not clear to the reviewer. Also, it should be "load-deflection" rather than "load-strain".

Response 5: Thank you very much for your comment. We have added the legend for curves in Figure 6 (Figure 7 in the previous version). The abscissa was “load-strain”, we have revised it.

Comment 6: line 231. A blank page. 

Response 6: Thank you very much for your comment.

Comment 7: Figure 8. It seems the result curves have been smoothed. That will deeply damage the credibility of the figure.  And the figure itself is not sufficient to show the "good predictability". A table with the experimental and simulated characteristic properties and the comparison will be much more convincing. 

Response 7: Thank you very much for your comment. Yes, we had smoothed the curves. In the revised version, we showed the no-smoothed curves.

Comment 8: Please also double-check the variable symbols. Some of them in the figures were not in italics.

Response 8: Thank you very much for your professional comment. We have double-checked the variables symbols.

Reviewer 3 Report

1. The article is devoted to investigations on the ductility performance of reinforced concrete columns under eccentric load. Column models are reinforced with steel bars with yield strength of 500 MPa. The authors should specify in the text whether they mean the characteristic or design strength. It seems to be characteristic. It is recommended to change the title of the article, since it is a reinforced concrete column, not a concrete one.

The authors give an overview of the state of research. However, this review is too superficial. Therefore, the scientific novelty of the research is not quite clear. Reinforcement with a characteristic yield strength of 500 MPa is widely used in construction. Moreover, studies on application of reinforcement with yield strength 600 MPa are known. Therefore, justification of the novelty of the research should be improved.

3. line 56: Which parameters do the authors have in mind. Please specify them.

4. Provide a stress - strain diagram for steel reinforcement.

5. Table 2: Is this data experimental?

6. The description of the test is too concise. Give a more detailed description, load graph. What were the parameters measured with?

7. Figure 7 is incorrect. 6000 mm = 6 m. This deflection of steel reinforcement is more than strange. I think the authors messed up with the units and the name of the axes.

8. The considered samples are very short. How to go from the results to the structures of real columns? I mean, the results do not take into account the effect of slenderness. Please give an explanation.

9. Should give a more detailed description of the numerical model, the method of numerical analysis, etc. Including figures.

10. The term in English needs to be verified. For example, "compression zone height". "Height" should be replaced with "depth". 

11. How do the authors define the relative depth of the compressed portion of the section? Give the formula.

12. Supplement the conclusions with quantitative data.

The term in English needs to be verified. For example, "compression zone height". "Height" should be replaced with "depth". It is recommended to change the title of the article, since it is a reinforced concrete column, not a concrete one.

Author Response

#3 Reviewer:

General comment: The article is devoted to investigations on the ductility performance of reinforced concrete columns under eccentric load. Column models are reinforced with steel bars with yield strength of 500 MPa.

Response: Thank you very much for your comments. We have revised the manuscript, the revised part was marked in red in the revised manuscript, and the point-by-point reply are shown below, please check.

Comment 1: The authors should specify in the text whether they mean the characteristic or design strength. It seems to be characteristic. It is recommended to change the title of the article, since it is a reinforced concrete column, not a concrete one.

Response 1: Thank you very much for your comments. We have specified the mean of the strength in the revised manuscript:

“In this paper, the ductility performance of six concrete columns with 500MPa grade (the yield strength was 500MPa) high strength steel reinforcement under eccentric loading was analyzed by the experiment and the numerical models.”

Please see Lines 70-71 in the revised manuscript for more details.

Comment 2: The authors give an overview of the state of research. However, this review is too superficial. Therefore, the scientific novelty of the research is not quite clear. Reinforcement with a characteristic yield strength of 500 MPa is widely used in construction. Moreover, studies on application of reinforcement with yield strength 600 MPa are known. Therefore, justification of the novelty of the research should be improved.

Response 2: Thank you very much for your comments. We have added more literature about the high-strength reinforcement in the introduction, and revised the justification of the novelty of the research:

High-strength reinforcement can reduce the amount of steel used [17]. There were lots of researches on the configuration of high-strength reinforced concrete members, in the works of Refs. [18–23], the bearing capacity and seismic performance of the concrete members with 500 MPa high-strength steel reinforcement were analyzed. In addition to 500MPa grade steel bars, some scholars were paying attention to 600MPa grade steel bars, for examples, Feng et al. [24] conducted experimental research on the seismic performance of reinforced concrete beam column joints with 600MPa grade steel reinforcement, and the results showed that the joints with 600MPa grade reinforcement have good seismic performance. Lee et al. [25] studied the performance of 600MPa plates and circular hollow section joints under external torque and shear forces through experiments and finite element analysis, and compared the results with current design equations. Lee et al. [26] studied the influences of microstructure and inclusion on cold cracking with 600 MPa grade steel welds. Shi et al. [27] conducted experimental and theoretical research on the seismic performance of concrete columns with 600MPa grade reinforcement on the condition of high axial compression ratios, and suggested setting the design strength of 600MPa grade steel reinforcements at 560MPa.

Researchers had paid a lot of attention to the bearing capacity and seismic performance of concrete member with high-strength steel reinforcement, with few people holding on to the ductility of eccentrically compressed components.

Please see Lines 46-69 in the revised manuscript for more details.

Comment 3: line 56: Which parameters do the authors have in mind. Please specify them.

Response 3: Thank you very much for your comments. We have specified the parameters in the revised manuscript: “The various influencing parameters including reinforcement ratio, eccentricity ratio and concrete strength on the ductility of the specimen were systematically studied…”

Please see Lines 72-74 in the revised manuscript for more details.

Comment 4. Provide a stress - strain diagram for steel reinforcement.

Response 4: Thank you very much for your comments. We have added the stress-strain diagram for steel reinforcement, please see figure 5.

Comment 5. Table 2: Is this data experimental?

Comment 5. Thank you for your comments. Yes, Table 2 shows the experiment data, we have revised the table name.

Comment 6. The description of the test is too concise. Give a more detailed description, load graph. What were the parameters measured with?

Response 6: Thank you very much for your comments. We have given a more detailed description of the test:

“The main measurement contents in the experiment include deflection, force, strain, etc. The horizontal deflections were measured by LVDTs, while the vertical deflections were measured by the instrument itself. The strains of the reinforcements were measured by using a bonded strain gauge. The layout situation of the LVDT and reinforcement strain gauge is shown in the Figure 4”.

Please see Lines 110-114 in the revised manuscript for more details.

Comment 7. Figure 7 is incorrect. 6000 mm = 6 m. This deflection of steel reinforcement is more than strange. I think the authors messed up with the units and the name of the axes.

Response 7: Thank you very much for your careful review. The horizontal axis was written incorrectly in the previous version, we have revised it. The revised horizontal coordinate name is “Load-strain”.

Comment 8. The considered samples are very short. How to go from the results to the structures of real columns? I mean, the results do not take into account the effect of slenderness. Please give an explanation.

Response 8: Thank you very much for your professional comments. Yes, the influence of slenderness ratio was ignored in the study, which was also a major deficiency of this study. The numerical simulation method used in this article was a cross-sectional analysis method, so the influence of column length could not be considered. We will pay more attention to this issue in future research.

We have explained this limitation in the revised manuscript: “and the disadvantage is that the influence of column length, i.e. slenderness ratio, cannot be considered.”

“There are some limitations in the work of this manuscript, for examples, the results were based on cross-sections and ignored the influence of slenderness ratio, and the bidirectional eccentricity was not considered. In future research, more attention will be paid to spatial effects.”

Please see Lines 411-414 in the revised manuscript for more details.

Comment 9. Should give a more detailed description of the numerical model, the method of numerical analysis, etc. Including figures.

Response 9: Thank you very much for your professional comments. We have added more details of the numerical mode: “…The sectional numerical models were established by using XTRACT [29] software, and the method adopted by the software is to fibrosis the section, as shown in Figure 9, then gradually increase the sectional curvature, and use the numerical integration method to obtain the corresponding moment, which has high calculation accuracy and efficiency, and the disadvantage is that the influence of column length, i.e. slenderness ratio, cannot be considered.”

Please see Lines 223-230 in the revised manuscript for more details.

Comment 10. The term in English needs to be verified. For example, "compression zone height". "Height" should be replaced with "depth". 

Response 10: Thank you very much for your professional comments. We have revised and checked the manuscript.

Comment 11. How do the authors define the relative depth of the compressed portion of the section? Give the formula.

Response 11: Thank you very much for your comments. We have added the formula of the relative depth of the compression portion of the section.

where ξ is the relative depth of compression zone, and it can be calculated by the following formula,

Please see Lines 372-376 in the revised manuscript for more details.

Comment 12. Supplement the conclusions with quantitative data.

Response 12: Thank you very much for your comments. We have supplemented the conclusions with quantitative data:

(4) The ductility of column section increases with the increase of eccentricity ratio, and the change is greater when the reinforcement ratio is smaller. For a smaller reinforcement ratio, when the eccentricity increases from 0.475 to 0.85, the section ductility increases by about 113%, and for a larger reinforcement ratio, the ductility increases by about 45%.

(5) The simplified calculation formula of curvature ductility can accurately evaluate the ductility of the concrete columns with high strength steel reinforcement under eccentric loading.

(6) The ductility of the concrete columns with high strength steel reinforcement under eccentric loading can be improved by improving the concrete strength and increasing eccentricity ratio. Increasing the reinforcement ratio has a negative impact on ductility, so when there is a demand for high ductility, it is necessary to control the reinforcement ratio to not exceed a certain limit value.”

Please see Lines 398-410 in the revised manuscript for more details.

Reviewer 4 Report

The authors investigate the ductility of reinforced concrete columns made with 500MPa grade reinforcement (the authors call it high-strength steel). The investigation includes an experimental part and a numerical part, the latter including a parametric analysis. The subject in relevant, the experimental study interesting but the paper as two main problems: extreme lack of clarity and the validity of the parametric study.

Major issues

page 2 line 52 :  "Moreover, there is still no method to quantitatively evaluate the ductility of 500 MPa reinforced concrete under eccentric compression." I am not sure what the authors mean. Calculation of the ductility of the structural members is included in most codes. For instance, the eurocode EN 1998-3, 2005, contains expressions for the computation of the chord rotation capacity of RC members (and these expressions apply to every steel grade).

p.8, section 4 : no information is given about the finite element model, analysis, etc. This is strange having in view the validation of the experimental results and the parametric analysis.

p.11 Figure 8 : experimental and numerical results for specimens 2, 3 and 6 are rather divergent. How do the authors explain this? Do the authors think that such a divergence is not problematic, at least for the parametric numerical analysis that is presented afterwards? This is rather upsetting because the ductility is the characteristic of the columns that this paper intends to investigate.

p.12, l.263 : "It can be seen that the bending moment bearing capacity of the specimen with a large eccentricity within the parameter setting range is generally greater than that of a specimen with a small eccentricity." This can be seen in figure 10? What is the "bending moment bearing capacity"? Do the authors mean the ductility?

p.13, 277 : "The possible reason is that the reinforcement ratio ρ = 1.23% is too small. " What do the authors mean?

p.13, 279 : "range of small eccentricity or boundary failure" What is the boundary failure?

p.14, 286 : "the strength of the concrete can be improved by improving the ductility of the cross-section of the eccentric compression column."???

p.14, 295 : "the specimen with a higher reinforcement ratio has a larger bending capacity" The main concepts should be better explained. What is the "bending capacity"? Are the authors referring to strength or to rotation capacity (or to both)? I think the authors mean the rotation capacity (and in fact I think it is correct to call bending capacity to the rotation capacity) the problem is that figure 12 shows the opposite of this statement (the ductility increases when the reinforcement ratio decreases), which makes everything very confusing.

p.14, l.300 : Referring to Figure 12, the authors write "The smaller the eccentricity is, the more obvious the horizontal section of the curve is after the bearing capacity reaches the limit bending moment". Either this is not correct or I do not understand what they mean.

p.14 and 15, Figure 12 : Wrong sub-captions (a=c and b=d)

p.15, l.308 : "the decrease of the concrete strength is faster than that of the concrete, and the decrease of the eccentricity is faster than that of the eccentricity"???

p.15, figure 13 : the identification of the curves is wrong

p15, l.319 : "It can be seen from the figure that the eccentricity has no significant effect on the bending capacity of the specimen section, and the bending resistance of the specimen under different parameters". Figure 14 does not show this.

p.16, figure 14d) : Are all the curves plotted?

p.16, l.335 : "Both increase with the increase of the eccentricity" Both? What are the authors referring to?

p.16, l.338 : "which may be due to the reinforcement ratio Too large results in insufficient increase of eccentricity" What do the authors mean?

p.17, Figure 15 : the results appear to be wrong... the ductility increases with the reinforcement ratio?

p.17 l.355  : the "relative compression zone height calculated at the limit state" is defined only in page 17, but was already employed in page 16...

p.17, l.356 : "It can be seen that Eq. (2) The trend"???

p.18, l.374 : "The ductility of the section of eccentrically compressed column can be improved by
increasing the concrete strength, controlling the maximum reinforcement ratio, and reducing the eccentricity." Improve the ductility by reducing the eccentricity? This conclusion appears to be wrong!

Minor issues

page 5, line 104 : "The vertical cracks of the strips ". Which strips?

p.5, l. 110 : "and each final specimen has a large deformation and deformation" Please clarify the sentence.

p.5, l.110 : "The bearing capacity was reduced to 85% of the ultimate load, that is, the deformation was more significant before the nominal limit displacement. " Can the authors explain what they mean?

p.6, l.145 : "and the test pieces were in the elastic stage and spanned the middle level". Spanned the middle level? What do the authors mean?

p.7, l.176 : " the eccentricity has a bearing capacity on the test piece. " Can the authors explain what they mean?

p.7, l.177 : "The larger the effect," Which effect?

p.8, Fig. 7 : the curves are not identified

p.9, l.246 : "they can be determined on the curve of moment- curvature by drawing approach". In fact, even though the procedure employed to determine the ultimate and yield curvatures can be represented by the scheme in Fig. 9, it is not a "drawing approach"!

p.12, l.258 : "the specimen with larger concrete strength becomes more obvious in the horizontal section after the bending moment reaches the ultimate bearing capacity" Can the authors please explain what they mean?

p13, l.278 and 281 : "The calculation of the bias voltage"???

p13, 281 : "Biased to eccentric damage. " Can the authors please explain what they mean?

English issues (suggestions)

page 1, abstract : "Numerical models were established, and the reliability of the model was verified." -> "Numerical models were established, and their reliability was verified.

page 1, line 39 : " is a typical ductility failure." -> " is a typical DUCTILE failure."

p.1, l.41 : "There are many researchers on the eccentric compression performance of columns" -> "There are many RESEARCHES on the eccentric compression performance of columns"

p.2, l.62 : "the length of columns are all 1700mm," -> "the length of columns IS 1700mm,"

p.5, l.103 : " the tensile steel bar began to yields, " -> " the tensile steel bar began to YIELD, "

p.5,  l.107 : "When the load was continued" -> "Under continuation of the loading"

p.5, l.118 : "It can be seen that the tensile strength of the tensile steel bars was crushed earlier than the concrete in the compression zone" -> "It can be seen that the tensile strength of the tensile steel bars was ATTAINED BEFORE the concrete in the compression zone CRUSHED"

p.6, l.126 : "The distance between several cracks was about 8 cm. "  -> "The distance between cracks was about 8 cm. "

p.6, l.136 : "The ductility of the test piece is weak, " -> "The ductility of the test piece is LOW, "

p.7, l.164 : "the load-deflection curve of the specimen with a large reinforcement ratio increased rapidly" -> "the DEFLECTION of the specimen with a large reinforcement ratio increased rapidly"

p.7, l.167 : "the load-displacement curve of the specimen decreases more gently" -> "the APPLIED LOAD decreases more gently"

p.7, l.178 : "has a smaller height of the concrete relative to the compression zone" -> "has a smaller compression zone"

p.8, l.191 : "The measured strain value of the obtained steel bar is" -> "The measured strain IN THE steel bar is"

p.8, l.208 : "to simulate the mechanical performance of 500MPa reinforced concrete column bias" Bias?

p.9, l.216 : parabplic -> parabolic

p.12, l.236 : "XTRACT was used for parameter analysis." -> "XTRACT was used IN A parametRIC analysis."

p.12, l.236 : "500MPa reinforced concrete" -> "500MPa STEEL REINFORCEMENT"

Author Response

#4 Reviewer:

General comment: The authors investigate the ductility of reinforced concrete columns made with 500MPa grade reinforcement (the authors call it high-strength steel). The investigation includes an experimental part and a numerical part, the latter including a parametric analysis. The subject in relevant, the experimental study interesting but the paper as two main problems: extreme lack of clarity and the validity of the parametric study.
Response: Thank you very much for your comments. We have revised the manuscript, the revised part was marked in red in the revised manuscript, and the point-by-point reply are shown below, please check.

Major issues

Comment 1: page 2 line 52 :  "Moreover, there is still no method to quantitatively evaluate the ductility of 500 MPa reinforced concrete under eccentric compression." I am not sure what the authors mean. Calculation of the ductility of the structural members is included in most codes. For instance, the eurocode EN 1998-3, 2005, contains expressions for the computation of the chord rotation capacity of RC members (and these expressions apply to every steel grade).

Response 1: Thank you very much for your comments. Yes, many specifications had established ductility calculation methods, but currently there were few methods for calculating the ductility capacity of concrete columns with high strength steel rebar under eccentric loading, based on this, we conducted this research:

“Researchers had paid a lot of attention to the bearing capacity and seismic performance of concrete member with high-strength steel reinforcement, with few people holding on to the ductility of eccentrically compressed components. The ductility of members under load is insufficiently studied, and the ductility problem has been emphasized in the eccentric compression of concrete columns. The Eurocode EN 1998-3, 2005, contains expressions for the computation of the chord rotation capacity of RC members, and this paper will be a supplement to the quantitative research on the ductility of high-strength reinforced concrete columns under eccentric compression.”

Please see Lines 62-69 in the revised manuscript for more details.

Comment 2: p.8, section 4 : no information is given about the finite element model, analysis, etc. This is strange having in view the validation of the experimental results and the parametric analysis.

Response 2: Thank you very much for your comments. The manuscript adopted a cross-sectional numerical analysis method, not the finite element method, which has been explained in the revised manuscript:

“The sectional numerical models were established by using XTRACT [29] software, and the method adopted by the software is to fibrosis the section, as shown in Figure 9, then gradually increase the sectional curvature, and use the numerical integration method to obtain the corresponding moment, which has high calculation accuracy and efficiency, and the disadvantage is that the influence of column length, i.e. slenderness ratio, cannot be considered.”

Please see Lines 223-230 in the revised manuscript for more details.

Comment 3: p.11 Figure 8 : experimental and numerical results for specimens 2, 3 and 6 are rather divergent. How do the authors explain this? Do the authors think that such a divergence is not problematic, at least for the parametric numerical analysis that is presented afterwards? This is rather upsetting because the ductility is the characteristic of the columns that this paper intends to investigate.

Response 3: Thank you very much for your comments. Due to the fact that the moment and curvature in the experiment were not directly obtained, there was data distortion during the process of obtaining curvature, which also leaded to unsatisfactory experimental data for several specimens. This point has been supplemented and explained in the revised manuscript:

“The moment and curvature of the specimen can not be directly obtained during loading, where the moment was obtained by the load and eccentricity, and the curvature was converted from the data collected by the strain gauge pasted on the reinforcement, and it can be written as,….”

“Since the curvatures were obtained through strain conversion, and strain testing is not stable, the experimental moment curvature curves of some specimens, such as HRCC-2 and HRCC-5, are not ideal. HRCC-3 experienced instrument failure during the loading process of the test, the curve after the bearing capacity decreased failed to captured. However, the test curve before reaching the ultimate bearing capacity was close to the numerical curve, and the numerical results could compensate for the shortcomings of the test. However, overall, the numerical results curves were consistent with the experimental results curves.”

Please see Lines 243-246 and Lines 251-258 in the revised manuscript for more details.

Comment 4: p.12, l.263 : "It can be seen that the bending moment bearing capacity of the specimen with a large eccentricity within the parameter setting range is generally greater than that of a specimen with a small eccentricity." This can be seen in figure 10? What is the "bending moment bearing capacity"? Do the authors mean the ductility?

Response 4: Thank you very much for your comments. In the previous manuscript, there were some description errors. We have revised the "bending moment bearing capacity" to “bear capacity”.

Comment 5: p.13, 277 : "The possible reason is that the reinforcement ratio ρ = 1.23% is too small. " What do the authors mean?

Response 5: Thank you very much for your comments. In the previous manuscript, there were some description errors. We have deleted the original words and rephrased them:

“This is because the reinforcement ratio was too large, resulting in the section being in a small eccentric compression failure state.”

Please see Lines 308-308 in the revised manuscript for more details.

Comment 6: p.13, 279 : "range of small eccentricity or boundary failure" What is the boundary failure?

Response 6: Thank you very much for your comments. In the previous manuscript, there were some description errors. We have deleted the original words and rephrased them:

“when the moments reach the ultimate bearing capacity, the curve of sections with high concrete strength is smoother. Within a certain range, the section of concrete columns with higher strength concrete exhibits better ductility.”

Please see Lines 295-297 in the revised manuscript for more details.

Comment 7: p.14, 286 : "the strength of the concrete can be improved by improving the ductility of the cross-section of the eccentric compression column."???

Response 7: Thank you very much for your comments. In the previous manuscript, there were some description errors. We have deleted the original words and rephrased them:

“This is because the reinforcement ratio was too large, resulting in the section being in a small eccentric compression failure state.

Overall, the strength of concrete had increased from 20MPa to 45MPa, with a ductility increase of -13% to 61%. It can be concluded that in the design of section of concrete columns with high strength steel reinforcement under eccentric loading, adjusting the strength of the concrete relative to adjusting the reinforcement ratio has no significant effect on the ductility of the section. In the case of proper reinforcement ratio and eccentricity, the ductility of sections can be improved by properly increasing the concrete strength.”

Please see Lines 309-315 and Lines 307-308 in the revised manuscript for more details.

Comment 8: p.14, 295 : "the specimen with a higher reinforcement ratio has a larger bending capacity" The main concepts should be better explained. What is the "bending capacity"? Are the authors referring to strength or to rotation capacity (or to both)? I think the authors mean the rotation capacity (and in fact I think it is correct to call bending capacity to the rotation capacity) the problem is that figure 12 shows the opposite of this statement (the ductility increases when the reinforcement ratio decreases), which makes everything very confusing.

Response 8: Thank you very much for your comments. This was aimed at ultimate bearing capacity. We have revised them.

Comment 9: p.14, l.300 : Referring to Figure 12, the authors write "The smaller the eccentricity is, the more obvious the horizontal section of the curve is after the bearing capacity reaches the limit bending moment". Either this is not correct or I do not understand what they mean.

Response 9: Thank you very much for your comments. In the previous manuscript, there were some description errors. We have deleted the original words and rephrased them:

“…the specimen with a higher reinforcement ratio had a larger bearing capacity. At the beginning of loading, the reinforcement ratio has no obvious effect on the stiffness of the specimen. After the edge of the tensile zone was cracked, the slopes of the moment-curvature curve of the section become smaller. Under the same action of the moment, the cross-section curvature of the specimen with a small reinforcement ratio is larger. When the section with smaller reinforcement ratio reaches the ultimate bearing capacity, its bending moment-curvature curve perform more gentle, which means that the sections with smaller reinforcement ratios have better ductility.”

Please see Lines 323-330 in the revised manuscript for more details.

Comment 10: p.14 and 15, Figure 12 : Wrong sub-captions (a=c and b=d)

Response 10: Thank you very much for your careful review. We have revised it in the revised version.

Comment 11: p.15, l.308 : "the decrease of the concrete strength is faster than that of the concrete, and the decrease of the eccentricity is faster than that of the eccentricity"???

Response 11: Thank you very much for your comments. In the previous manuscript, there were some description errors. We have deleted the original words and rephrased them:

“…and for sections with higher concrete strength, the reduction amplitude was larger. It can also be known that when the other conditions were the same, the ductility of the sections with higher concrete strength was generally larger, and the ductility of the sections with larger eccentricity was generally larger, for small eccentricity, when the reinforcement ratio increases from 0.34% to 1.23%, the ductility decreases by about 58%, and for large eccentricity, the ductility decreases by about 70%.”

Please see Lines 336-342 in the revised manuscript for more details.

Comment 12: p.15, figure 13 : the identification of the curves is wrong.

Response 12: Thank you very much for your careful review. We have revised it in the revised version.

Comment 13: p15, l.319 : "It can be seen from the figure that the eccentricity has no significant effect on the bending capacity of the specimen section, and the bending resistance of the specimen under different parameters". Figure 14 does not show this.

Response 13: Thank you very much for your comments. In the previous manuscript, there were some description errors. We have deleted the original words and rephrased them:

“…Before the concrete cracking, the eccentricity has little effect on the section stiffness, and after the concrete cracking, the slope of the section moment-curvature curve decreases, and the reduction amplitude increases with the increase of the eccentricity. After reaching the ultimate bearing capacity, the greater the section eccentricity, the smoother the bending moment curvature curve and the better the ductility.”

Please see Lines 347-352 in the revised manuscript for more details.

Comment 14: p.16, figure 14d) : Are all the curves plotted?

Response 14: Thank you very much for your careful review. We have revised it in the revised version.

Comment 15: p.16, l.335 : "Both increase with the increase of the eccentricity" Both? What are the authors referring to?

Response 15: Thank you very much for your careful review. “Both” means “Both line”. We have rewritten the description:

“It can be seen that the ductility of the column section is more sensitive to the reinforcement ratio than to concrete. In addition, In the four cases, the section ductility increases with the increase of eccentricity, the change is greater when the reinforcement ratio is smaller. For a smaller reinforcement ratio, when the eccentricity increases from 0.475 to 0.85, the section ductility increases by about 113%, and for a larger reinforcement ratio, the ductility increases by about 45%.”

Please see Lines 356-360 in the revised manuscript for more details.

Comment 16: p.16, l.338 : "which may be due to the reinforcement ratio Too large results in insufficient increase of eccentricity" What do the authors mean?

Response 16: Thank you very much for your comment. We have rewritten the description:

“It can be seen that the ductility of the column section is more sensitive to the reinforcement ratio than to concrete. In addition, In the four cases, the section ductility increases with the increase of eccentricity, the change is greater when the reinforcement ratio is smaller. For a smaller reinforcement ratio, when the eccentricity increases from 0.475 to 0.85, the section ductility increases by about 113%, and for a larger reinforcement ratio, the ductility increases by about 45%.”

Please see Lines 356-360 in the revised manuscript for more details.

Comment 17: p.17, Figure 15 : the results appear to be wrong... the ductility increases with the reinforcement ratio?

Response 17: Thank you very much for your comment. There was a label error in this Figure. We have revised it and rewritten the description:

“It can be seen that the ductility of the column section is more sensitive to the reinforcement ratio than to concrete. In addition, In the four cases, the section ductility increases with the increase of eccentricity, the change is greater when the reinforcement ratio is smaller. For a smaller reinforcement ratio, when the eccentricity increases from 0.475 to 0.85, the section ductility increases by about 113%, and for a larger reinforcement ratio, the ductility increases by about 45%.”

Please see Lines 356-360 in the revised manuscript for more details.

Comment 18: p.17 l.355  : the "relative compression zone height calculated at the limit state" is defined only in page 17, but was already employed in page 16...

Response 18: Thank you very much for your comment. We have removed the relevant description before the definition.

Comment 19: p.17, l.356 : "It can be seen that Eq. (2) The trend"???

Response 19: Thank you very much for your comment. We have revised it:

“The calculation result of the regression formula is shown in Figure 18. By using this fitting equation, the ductility coefficient of concrete columns with high strength steel reinforcement under eccentric loading can be quickly obtained.

Please see Lines 377-379 in the revised manuscript for more details.

Comment 20: p.18, l.374 : "The ductility of the section of eccentrically compressed column can be improved by increasing the concrete strength, controlling the maximum reinforcement ratio, and reducing the eccentricity." Improve the ductility by reducing the eccentricity? This conclusion appears to be wrong!

Response 20: Thank you very much for your comment. We have rewritten the conclusions:

“(3) The ductility of column section decreases with the increase of reinforcement ratio, especially when the eccentricity is large,decreases with the increase of reinforcement ratio, and the ductility of the sections with larger eccentricity was generally larger, for small eccentricity, when the reinforcement ratio increases from 0.34% to 1.23%, the ductility decreases by about 58%, and for large eccentricity, the ductility decreases by about 70%.

(4) The ductility of column section increases with the increase of eccentricity ratio, and the change is greater when the reinforcement ratio is smaller. For a smaller reinforcement ratio, when the eccentricity increases from 0.475 to 0.85, the section ductility increases by about 113%, and for a larger reinforcement ratio, the ductility increases by about 45%.

(5) The simplified calculation formula of curvature ductility can accurately evaluate the ductility of the concrete columns with high strength steel reinforcement under eccentric loading.

(6) The ductility of the concrete columns with high strength steel reinforcement under eccentric loading can be improved by improving the concrete strength and increasing eccentricity ratio. Increasing the reinforcement ratio has a negative impact on ductility, so when there is a demand for high ductility, it is necessary to control the reinforcement ratio to not exceed a certain limit value.

Minor issues

Comment 21: page 5, line 104 : "The vertical cracks of the strips ". Which strips?

Response 21: Thanks for comment. We have rewritten the sentence: “Subsequently, vertical cracks appeared, and the concrete was crushed…”

Comment 22: p.5, l. 110 : "and each final specimen has a large deformation and deformation" Please clarify the sentence.

Response 22: Thanks for comment. We have rewritten the sentence:

“…Subsequently, there was the second stage, and the displacement control was adopted, and each step was 1mm of this stage. The load would be stopped when the load drops to 65% of the ultimate bearing capacity.”

Please see Lines 106-109 in the revised manuscript for more details.

Comment 23: p.5, l.110 : "The bearing capacity was reduced to 85% of the ultimate load, that is, the deformation was more significant before the nominal limit displacement. " Can the authors explain what they mean?

Response 23: Thanks for comment. Our original intention was that after the bearing capacity decreased to 65% of the ultimate bearing capacity, the experimental data was sufficient for us to obtain the ultimate displacement. We have rewritten the sentence:

“…Subsequently, there was the second stage, and the displacement control was adopted, and each step was 1mm of this stage. The load would be stopped when the load drops to 65% of the ultimate bearing capacity.”

Please see Lines 106-109 in the revised manuscript for more details.

Comment 24: p.6, l.145 : "and the test pieces were in the elastic stage and spanned the middle level". Spanned the middle level? What do the authors mean?

Response 24: Thanks for comment. We have revised the sentences: “At the beginning of loading of each component, the load was small, no cracks appeared in the test specimens, the deflection and the strain of the steel bar and concrete increased linearly with the load, and the specimens showed elastic deflection characteristics. As the load increases, cracks began to appear on the tensile side of the specimens and begin to widen and extend.”

Please see Lines 126-129 in the revised manuscript for more details.

Comment 25: p.7, l.176 : " the eccentricity has a bearing capacity on the test piece. " Can the authors explain what they mean?

Response 25: Thanks for comment. We have revised the description: “It can be seen that within the range of the experiment parameters, the eccentricity ratio had a great influence on the specimens, and the bearing capacity decreases with the increase of the eccentricity ratio.”

Please see Lines 194-196 in the revised manuscript for more details.

Comment 26: p.8, Fig. 7 : the curves are not identified.

Response 26: Thanks for comment. We have revised it.

Comment 27: p.9, l.246 : "they can be determined on the curve of moment- curvature by drawing approach". In fact, even though the procedure employed to determine the ultimate and yield curvatures can be represented by the scheme in Fig. 9, it is not a "drawing approach"!

Response 27: Thanks for comment. We have revised the description: “In this paper, the yield curvature was determined by the method shown in Figure 10: firstly, using the initial stiffness as an auxiliary line OA, and line OA intersects with the horizontal line of the ultimate load at point A; secondly, creating a vertical line AB, and line AB intersects with the bending moment curvature curve at point C; thirdly, connectting point O and point C, and extending it, and the line intersects with the horizontal line of the ultimate load at point D; lastly, the curvature corresponding to point D is the yield curvature.”

Please see Lines 280-285 in the revised manuscript for more details.

Comment 29: p.12, l.258 : "the specimen with larger concrete strength becomes more obvious in the horizontal section after the bending moment reaches the ultimate bearing capacity" Can the authors please explain what they mean?

Response 29: Thanks for comment. We have revised the description: “when the moment reaches the ultimate bearing capacity, the curve of sections with high concrete strength is smoother. Within a certain range, the section of concrete columns with higher strength concrete exhibits better ductility.”

Please see Lines 295-297 in the revised manuscript for more details.

Comment 30: p13, l.278 and 281 : "The calculation of the bias voltage"???

Comment 31: p13, 281 : "Biased to eccentric damage. " Can the authors please explain what they mean?

English issues (suggestions)

Comment 32: page 1, abstract : "Numerical models were established, and the reliability of the model was verified." -> "Numerical models were established, and their reliability was verified.

Response 32: Thanks for comment. We have revised it.

Comment 33: page 1, line 39 : " is a typical ductility failure." -> " is a typical DUCTILE failure."
Response 33: Thanks for comment. We have revised it.

Comment 34: p.1, l.41 : "There are many researchers on the eccentric compression performance of columns" -> "There are many RESEARCHES on the eccentric compression performance of columns"

Response 34: Thanks for comment. We have revised it.

Comment 35: p.2, l.62 : "the length of columns are all 1700mm," -> "the length of columns IS 1700mm,"

Response 35: Thanks for comment. We have revised it.

Comment 36: p.5, l.103 : " the tensile steel bar began to yields, " -> " the tensile steel bar began to YIELD, "

Response 36: Thanks for comment. We have revised it.

Comment 37: p.5,  l.107 : "When the load was continued" -> "Under continuation of the loading"

Response 37: Thanks for comment. We have revised it.

Comment 38: p.5, l.118 : "It can be seen that the tensile strength of the tensile steel bars was crushed earlier than the concrete in the compression zone" -> "It can be seen that the tensile strength of the tensile steel bars was ATTAINED BEFORE the concrete in the compression zone CRUSHED"

Response 38: Thanks for comment. We have revised it.

Comment 39: p.6, l.126 : "The distance between several cracks was about 8 cm. "  -> "The distance between cracks was about 8 cm. "

Response 39: Thanks for comment. We have revised it.

Comment 40: p.6, l.136 : "The ductility of the test piece is weak, " -> "The ductility of the test piece is LOW, "

Response 40: Thanks for comment. We have revised it.

Comment 41: p.7, l.164 : "the load-deflection curve of the specimen with a large reinforcement ratio increased rapidly" -> "the DEFLECTION of the specimen with a large reinforcement ratio increased rapidly"

Response 41: Thanks for comment. We have revised it.

Comment 42: p.7, l.167 : "the load-displacement curve of the specimen decreases more gently" -> "the APPLIED LOAD decreases more gently"

Response 42: Thanks for comment. We have revised it.

Comment 43: p.7, l.178 : "has a smaller height of the concrete relative to the compression zone" -> "has a smaller compression zone"

Response 43: Thanks for comment. We have revised it.

Comment 44: p.8, l.191 : "The measured strain value of the obtained steel bar is" -> "The measured strain IN THE steel bar is"

Response 44: Thanks for comment. We have revised it.

Comment 45: p.8, l.208 : "to simulate the mechanical performance of 500MPa reinforced concrete column bias" Bias?

Response 45: Thanks for comment. We have revised it: “Due to limited experimental research, and in order to analyze the ductility capacity of concrete columns with high strength steel reinforcement under eccentric loading more systematically.”

Comment 46: p.9, l.216 : parabplic -> parabolic

Response 46: Thanks for comment. We have revised it:

Comment 47 :p.12, l.236 : "XTRACT was used for parameter analysis." -> "XTRACT was used IN A parametRIC analysis."

Response 47: Thanks for comment. We have revised it:

Comment 48: p.12, l.236 : "500MPa reinforced concrete" -> "500MPa STEEL REINFORCEMENT"

Response 48: Thanks for comment. We have revised it:

Round 2

Reviewer 1 Report

In general, authors provided satisfactory comments,

Just one exception: 

* Authors provided clear response to comment 9 of reviewer, but their comment is still not complete because provided equation still have unknown parameter "x" i.e., depth of compression zone.

Authors should also explain how readers can calculate this parameter?

Just reminder: Ref. [31] in the manuscript is capable of providing this parameter which can help authors to provide comment about this issue.

Paper still needs Moderate editing of English language. Just one simple example in lines 225-227 "Due to limited experimental research, and in order to analyze the ductility capacity of concrete columns with high strength steel reinforcement under eccentric loading more systematically" This word is not complete and it haslack of meaning. It can be replaced as;

"Due to limited experimental research, numerical models were created to make systematic evaluations for concrete columns with high strength steel reinforcement under eccentric loading."

Please re-check and revise english language check in whole manuscript! I expect that this issues can be solved in proofreading process or "English editing" service of the Publisher.

Author Response

#1 Reviewer:

General comment: In general, authors provided satisfactory comments.

Response: Thank you very much for your comments. We have revised the manuscript, the revised part was marked in red in the revised manuscript, and the point-by-point reply are shown below, please check.

Comment 1: * Authors provided clear response to comment 9 of reviewer, but their comment is still not complete because provided equation still have unknown parameter "x" i.e., depth of compression zone.

Authors should also explain how readers can calculate this parameter?

Just reminder: Ref. [31] in the manuscript is capable of providing this parameter which can help authors to provide comment about this issue.

Response 1: Thank you for your review again. We have added more information of parameters ‘x’, please see Lines 375-384 for more details:

 ” where  denotes the effective depth of section; x denotes the depth of compression zone, and when the section is subjected to large eccentric compression failure, it can be calculated by the following formula,

where  is a constant coefficient,  denotes strength of concrete,  is the width of section,  and  are the yield strength of compression and tension reinforcement respectively,  and  denote sectional area of compression and tension reinforcement respectively,e is the eccentricity.

When the section is subjected to small eccentric compression failure, x can be calculated by the following formula,

where  denotes the stress of tension reinforcement. More details can be found in the code GB50010-2010 [26].”

Comment 2: Paper still needs Moderate editing of English language. Just one simple example in lines 225-227 "Due to limited experimental research, and in order to analyze the ductility capacity of concrete columns with high strength steel reinforcement under eccentric loading more systematically" This word is not complete and it hasl ack of meaning. It can be replaced as;

"Due to limited experimental research, numerical models were created to make systematic evaluations for concrete columns with high strength steel reinforcement under eccentric loading."

Please re-check and revise english language check in whole manuscript! I expect that this issues can be solved in proofreading process or "English editing" service of the Publisher.

Response 2: Thank you very much for your careful review. We have revised the manuscript, and double checked the revised manuscript.

Reviewer 3 Report

The authors did a good job and carefully considered my comments. I believe that the article can be accepted in its current form.

Author Response

#3 Reviewer:

The authors did a good job and carefully considered my comments. I believe that the article can be accepted in its current form.

The manuscript presents an investigation of the ductility of reinforced concrete columns made with 500MPa grade reinforcement (the authors call it high-strength steel). The investigation includes an experimental part and a numerical part, the latter including a parametric analysis. The subject is relevant, the experimental study interesting and well complemented by the parametric study. And it was much improved since the original proposal.

Response: We sincerely appreciate your help.

Reviewer 4 Report

The manuscript presents an investigation of the ductility of reinforced concrete columns made with 500MPa grade reinforcement (the authors call it high-strength steel). The investigation includes an experimental part and a numerical part, the latter including a parametric analysis. The subject is relevant, the experimental study interesting and well complemented by the parametric study. And it was much improved since the original proposal.

I will only mention a few very minor points:

- References 25 and 26 are rather strange... They are not about reinforced concrete elements.

page 2 , line 64 : "Researchers had paid a lot of attention" -> "Researchers HAVE paid a lot of attention"

p.2, line 76 : "and a simplified calculation formula [...] was proposed" -> "and a simplified calculation formula [...] IS proposed"

p.4, l.112 : "The main measurement contents in the experiment include deflection, force, strain,
etc." Etc.? What else was measured?

p.6, l.158 : "are knowshown in Figure 6" -> "are shown in Figure 6"

p.9, l.228 : "the method adopted by the software is to fibrosis the section" Fibrosis?

p.11, l.265 : "XTRACT was in a parameter analysis. " -> "XTRACT was USED in a parameter analysis."

p.12, l.298  : "Within a certain range, the section of concrete columns with higher strength concrete exhibits better ductility." -> "Within a certain range, the concrete columns with higher strength concrete exhibit better ductility."

p.12, l.312 : "ductility increase of -13% to 61%" . Increase of -13%?

p.17, l.395 : "The ductility of column section decreases with the increase of reinforcement ratio,
especially when the eccentricity is large,decreases with the increase of reinforcement ratio". Repetition of "decreases with the increase of reinforcement ratio"

The English Language is much better now. I added some suggestions in the Comments section.

Author Response

#4 Reviewer:

The manuscript presents an investigation of the ductility of reinforced concrete columns made with 500MPa grade reinforcement (the authors call it high-strength steel). The investigation includes an experimental part and a numerical part, the latter including a parametric analysis. The subject is relevant, the experimental study interesting and well complemented by the parametric study. And it was much improved since the original proposal.

Response: Thank you very much for your careful review. We have revised the manuscript, the revised part was marked in red in the revised manuscript, and the point-by-point reply are shown below, please check.

Comment 1: - References 25 and 26 are rather strange... They are not about reinforced concrete elements.

Response 1: Thanks for comment. We have checked the literature again and it is true that these two references are about high-strength steel, so we have deleted them.

Comment 2: page 2 , line 64 : "Researchers had paid a lot of attention" -> "Researchers HAVE paid a lot of attention"

Response 2: Thanks for comment. We have revised it. Please see Lines 60.

Comment 3: p.2, line 76 : "and a simplified calculation formula [...] was proposed" -> "and a simplified calculation formula [...] IS proposed"

Response 3: Thanks for comment. We have revised it. Please see Lines 74.

Comment 4: p.4, l.112 : "The main measurement contents in the experiment include deflection, force, strain,
etc." Etc.? What else was measured?

Response 4: Thanks for comment. We have deleted it. Please see Lines 108.

Comment 5: p.6, l.158 : "are knowshown in Figure 6" -> "are shown in Figure 6"

Response 5: Thanks for comment. We have revised it. Please see Lines 154.

Comment 6: p.9, l.228 : "the method adopted by the software is to fibrosis the section" Fibrosis?

Response 6: Thanks for comment. We have revised it. Please see Lines 255: ”… and the method adopted by the software is to mesh the section…”

Comment 7: p.11, l.265 : "XTRACT was in a parameter analysis. " -> "XTRACT was USED in a parameter analysis."

Response 7: Thanks for comment. We have revised it. Please see Lines 263.

Comment 8: p.12, l.298  : "Within a certain range, the section of concrete columns with higher strength concrete exhibits better ductility." -> "Within a certain range, the concrete columns with higher strength concrete exhibit better ductility."

Response 8: Thanks for comment. We have revised it. Please see Lines 296.

Comment 9: p.12, l.312 : "ductility increase of -13% to 61%" . Increase of -13%?

Response 9: Thanks for comment. We have revised it. Please see Lines 309: “…the ductility of cases exhibits varying degrees of increase, in which the maximum growth rate reached 61%...”

Comment 10: p.17, l.395 : "The ductility of column section decreases with the increase of reinforcement ratio,
especially when the eccentricity is large,decreases with the increase of reinforcement ratio". Repetition of "decreases with the increase of reinforcement ratio"

Response 10: Thanks for comment. We have revised it. Please see Lines 402: “The ductility of column section decreases with the increase of reinforcement ratio, especially when the eccentricity is large”